# A tumor-targeted trimeric 4-1BB-agonistic antibody induces potent anti-tumor immunity without systemic toxicity

Marta Compte[1], Seandean Lykke Harwood[2], Ines G. Muñoz[3], Rocio Navarro[4], Manuela Zonca[1], Gema Perez-Chacon[5,6], Ainhoa Erce-Llamazares[1], Nekane Merino [7], Antonio Tapia-Galisteo[4], Angel M. Cuesta[4], Kasper Mikkelsen[2], Eduardo Caleiras[8], Natalia Nuñez-Prado[4], M. Angela Aznar[9], Simon Lykkemark[2], Jorge Martínez-Torrecuadrada[3], Ignacio Melero[9,10,11,12], Francisco J. Blanco [7,13], Jorge Bernardino de la Serna [14,15], Juan M. Zapata[5,6], Laura Sanz [4] & Luis Alvarez-Vallina[2,16,17]

The costimulation of immune cells using first-generation anti-4-1BB monoclonal antibodies (mAbs) has demonstrated anti-tumor activity in human trials. Further clinical development, however, is restricted by significant off-tumor toxicities associated with FcγR interactions. Here, we have designed an Fc-free tumor-targeted 4-1BB-agonistic trimerbody, 1D8$^{N/C}$EGa1, consisting of three anti-4-1BB single-chain variable fragments and three anti-EGFR single-domain antibodies positioned in an extended hexagonal conformation around the collagen XVIII homotrimerization domain. The1D8$^{N/C}$EGa1 trimerbody demonstrated high-avidity binding to 4-1BB and EGFR and a potent in vitro costimulatory capacity in the presence of EGFR. The trimerbody rapidly accumulates in EGFR-positive tumors and exhibits anti-tumor activity similar to IgG-based 4-1BB-agonistic mAbs. Importantly, treatment with 1D8$^{N/C}$EGa1 does not induce systemic inflammatory cytokine production or hepatotoxicity associated with IgG-based 4-1BB agonists. These results implicate FcγR interactions in the 4-1BB-agonist-associated immune abnormalities, and promote the use of the non-canonical antibody presented in this work for safe and effective costimulatory strategies in cancer immunotherapy.

---

[1] Department of Antibody Engineering, Leadartis SL, 28008 Madrid, Spain. [2] Immunotherapy and Cell Engineering Laboratory, Department of Engineering, Aarhus University, 8000C Aarhus, Denmark. [3] Crystallography and Protein Engineering Unit, Spanish National Cancer Research Centre (CNIO), 28029 Madrid, Spain. [4] Molecular Immunology Unit, Hospital Universitario Puerta de Hierro Majadahonda, 28222 Majadahonda, Madrid, Spain. [5] Instituto de Investigaciones Biomédicas Alberto Sols (IIBm), CSIC-UAM, 28029 Madrid, Spain. [6] Instituto de Investigación Sanitaria La Paz (IdiPaz), 28029 Madrid, Spain. [7] Structural Biology Unit, CIC bioGUNE, Parque Tecnológico de Bizkaia, 48160 Derio, Spain. [8] Histopathology Unit, Spanish National Cancer Research Centre (CNIO), 28029 Madrid, Spain. [9] Department of Immunology and Immunotherapy, Center for Applied Medical Research (CIMA), University of Navarra, 31008 Pamplona, Spain. [10] Department of Immunology, University Clinic, University of Navarra, 31008 Pamplona, Spain. [11] Instituto de Investigación Sanitaria de Navarra (IdISNA), 31008 Pamplona, Spain. [12] CIBERONC-Centro virtual de Investigación Biomédica en red de Oncología, 28029 Madrid, Spain. [13] IKERBASQUE, Basque Foundation for Science, 48013 Bilbao, Spain. [14] Central Laser Facility, Science and Technology Facilities Council, Rutherford Appleton Laboratory, Research Complex at Harwell, OX11 0QX Harwell-Oxford, UK. [15] Department of Physics, King's College London, WC2R 2LS London, UK. [16] Cancer Immunotherapy Unit (UNICA), Department of Immunology, Hospital Universitario 12 de Octubre, 28041 Madrid, Spain. [17] Immuno-Oncology and Immunotherapy Group, Instituto de Investigación Sanitaria 12 de Octubre (i+12), 28041 Madrid, Spain. Correspondence and requests for materials should be addressed to L.A.-V. (email: lav@eng.au.dk)

Modulating immune responses using monoclonal antibodies (mAbs) is a promising approach to cancer therapy. Antagonistic mAbs directed against checkpoint inhibitors such as cytotoxic T-lymphocyte–associated antigen 4 and programmed cell death 1/programmed cell death ligand 1 (PD-L1) have been clinically approved, and agonistic mAbs targeting costimulatory receptors are undergoing clinical trials[1]. Costimulatory receptors of the tumor necrosis factor (TNF) receptor superfamily (TNFRSF), such as CD40, OX40, and 4-1BB, are particularly interesting targets, as these receptors are not constitutively expressed on resting naive T cells but acquired upon activation[2–4]. This limits the potential deleterious side effects of the treatment[5].

4-1BB (CD137, TNFRSF9) has only one confirmed ligand [4-1BB-Ligand (4-1BBL), TNFSF9], which is expressed on macrophages, activated B cells, and dendritic cells[6]. Engagement of 4-1BB by its ligand or an agonistic antibody promotes T cell proliferation, cytokine production, and cytolytic effector functions and protects lymphocytes from programmed cell death[7,8]. Furthermore, engagement of 4-1BB on natural killer cells enhances cytokine release (including interferon (IFN)-γ)[9] and antibody-dependent cellular cytotoxicity[10,11]. Indeed, treatment of mice with 4-1BB-agonistic mAbs was found to induce tumor regression of established and poorly immunogenic tumors as early as 1997[12]. Since then, a large body of accumulated preclinical data has been gathered that supports the induction of 4-1BB signaling in cancer immunotherapy, both as a single agent and in combination therapies[13].

The effect of 4-1BB-agonistic mAbs is not spatially restricted to the tumor, and peripheral toxicities can therefore reduce the therapeutic window for 4-1BB-targeting therapies. In mice, 4-1BB mAbs have been shown to cause immune anomalies, notably polyclonal activation of CD8[+] T cells and secretion of inflammatory cytokines, which affected the function of liver, spleen, and bone marrow[14,15]. In clinical studies, an anti-4-1BB mAb (BMS-663513, urelumab) showed tolerable side effects in an initial Phase I trial, but a follow-up Phase II trial revealed severe liver toxicity in ≈10% of the patients that resulted in two fatalities[16]. As a consequence, trials with urelumab were terminated[17]. Recently, data were presented on a dose-escalation study with urelumab as monotherapy and in combination with nivolumab[18]. The reduced dose ameliorated liver toxicity; however, the clinical activity of urelumab at the tolerated dose was limited. An integrated safety analysis of patients treated with urelumab confirmed a clear association between transaminitis and urelumab dose[19]. Utomilumab is another anti-41BB mAb in clinical trials with a better safety profile than urelumab but is a relatively less potent 4-1BB agonist[20].

As it stands, costimulation by 4-1BB-agonistic mAbs is an otherwise viable therapeutic approach held back by off-tumor toxicities and could therefore benefit greatly from the addition of tumor-targeting functionality to restrict its effect to the tumor deposits. Furthermore, if this is conveyed by binding domains specific to cell surface tumor-associated antigens (TAAs), the anti-4-1BB antibodies will then cluster on the surface of cancer cells. This may allow the antibodies to mimic physiological 4-1BBL and could have a major impact on the induction of 4-1BB signaling. Importantly, 4-1BBL is a trimeric membrane protein and can be proteolytically processed into soluble trimeric ligands with a significantly reduced signaling activity compared to their transmembrane counterparts[21]. Signaling can be restored by higher-order oligomerization[21,22], cell surface display of anti-4-1BB single chain antibody fragments (scFv) expressed by tumor cells in fusion with membrane proteins[23,24], or antibody-mediated display by fusing the extracellular domain of 4-1BBL to a TAA-specific scFv[25]. Another strategy is the use of anti-4-1BB oligonucleotide aptamers instead of 4-1BBL[26,27]. In animal models, systemic delivery of a 4-1BB-agonistic aptamer conjugated to a prostate-specific membrane antigen aptamer led to superior therapeutic effect compared to immunoglobulin G (IgG)-based 4-1BB-agonistic antibodies[26]. It has also been recently reported that anchoring anti-4-1BB F(ab′)2 fragments and interleukin (IL)-2 on the surface of liposomes induced effective antitumor immunity without systemic toxicity[28].

In this article, we describe the adaptation of the first-generation 4-1BB agonistic IgG 1D8 to a recombinant antibody format, the trimerbody. This format is based on the fusion of antibody-derived binding domains to the small homotrimerization region from murine collagen XVIII, which yields trimeric antibodies[29–31]. Trimerbodies have two major advantages compared to the IgG mAb: they lack the fragment crystallizable (Fc) region involved in 4-1BB-mediated toxicity[20]; and they are trimeric, as is physiological 4-1BBL. We generated a panel of 1D8 scFv-based N-terminal trimerbodies (1D8[N]) using linkers of different lengths and identified 1D8[N18], the trimerbody with the longest linker, as showing the most potent costimulatory activity. 1D8[N18] was therefore used as the basis for a bispecific tumor-targeted trimerbody, 1D8[N/C]EGa1, by adding the epidermal growth factor receptor (EGFR)-binding EGa1 single-domain antibody[32]. Compared to both 1D8 IgG and 1D8[N18], 1D8[N/C]EGa1 was a more potent costimulator in vitro and showed enhanced tumor homing and tumor inhibition in vivo, with no indication of the 4-1BB mAb-associated toxicity.

## Results

**Design of 4-1BB-agonistic trimerbodies.** Using the scFv derived from the rat IgG2a, anti-4-1BB 1D8 mAb[33] (Fig. 1a), we designed a panel of 1D8 scFv-based N-terminal trimerbodies (1D8[N]). Three candidates were generated with varied lengths of the flexible linker connecting the 1D8 scFv to a murine collagen XVIII-derived homotrimerization (TIE[XVIII]) domain: 1D8[N0] has no linker, while 1D8[N5] and 1D8[N18] have 5- and 18-residue-long linkers, respectively (Fig. 1b). All three constructs were expressed by transfected HEK293 cells at similar levels to MFE-23[N18], a benchmark N-terminal trimerbody based on the anti-CEA MFE-23 scFv[29]. In western blot analysis under reducing conditions, the 1D8[N] trimerbodies were single-chain-type molecules with a migration pattern consistent with the molecular weights calculated from their amino acid sequences (34.4, 34.7, and 36.8 kDa, in the order of increasing linker length) (Supplementary Figure 1a). Additionally, they specifically recognized murine 4-1BB in fusion with human Fc (m4-1BB) immobilized on plastic, as determined by enzyme-linked immunosorbent assay (ELISA; Supplementary Figure 1b).

The three 4-1BB-specific trimerbodies were produced in stably transfected HEK293 cells and purified by immobilized metal affinity chromatography, with a yield of roughly 1 mg/l of conditioned medium (Supplementary Figure 2a). Their binding kinetics were then studied using biolayer interferometry (BLI). While all 1D8 antibodies showed low picomolar-range $K_D$ toward m4-1BB immobilized on biosensors, the 1D8[N] trimerbodies were found to dissociate at approximately half the rate of the parental 1D8 mAb (Fig. 2a, Supplementary Table 1). This indicates that the 1D8[N] trimerbodies are in fact functionally trivalent and therefore have a higher functional affinity than the bivalent 1D8 IgG. Furthermore, 1D8[N18], 1D8[N5], and 1D8[N0] showed extremely similar kinetic rate constants, indicating that none of the linker length variations structurally compromise the 1D8 scFv or sterically hinder its access to antigen. The ability of 1D8[N] trimerbodies to detect m4-1BB in a cellular context was studied by flow cytometry. All 1D8[N] trimerbodies bound to HEK293 cells

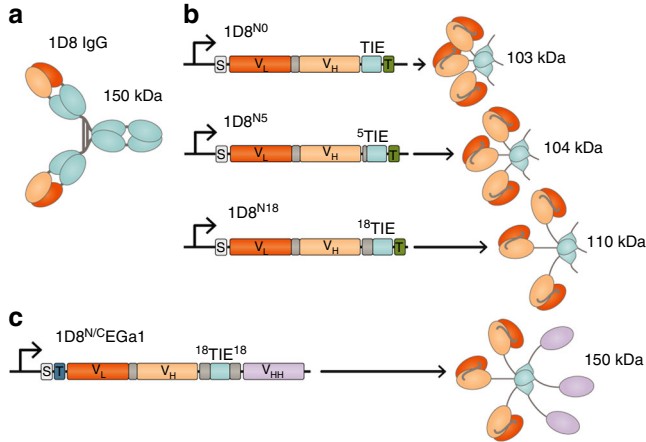

**Fig. 1** Schematic diagrams of 4-1BB-agonistic trimerbodies. Protein structure of the anti-4-1BB IgG (**a**) and the gene layout (left) and protein structure (right) of monospecific (**b**) and bispecific trimerbodies (**c**). The variable regions derived from 1D8 antibody are represented in orange, the anti-EGFR $V_{HH}$ EGa1 in violet, the structural domains in light blue, and the linker regions in gray. The 1D8 scFv-based N-terminal trimerbodies' (1D8$^N$) gene constructs (**b**) contain a signal peptide from oncostatin M (white box) and the 1D8 scFv gene ($V_L$-$V_H$) connected directly or through flexible linkers to the mouse TIE$^{XVIII}$ domain. In the bispecific 1D8$^{N/C}$EGa1 trimerbody (**c**), the anti-human EGFR $V_{HH}$ EGa1 is fused to the C-terminus of 1D8$^{N18}$ through a flexible linker. Arrows indicate the direction of transcription. His6-myc tag (green box) and FLAG-strep tags (dark blue box) were appended for immunodetection

transfected to express murine 4-1BB on their cell surface (HEK293$^{m4-1BB}$) but not to untransfected HEK293 cells (Supplementary Figure 2b). The binding of 1D8$^{N0}$ to HEK293$^{m4-1BB}$ cells was less efficient than that of the 1D8$^{N5}$ and 1D8$^{N18}$ (Supplementary Figure 2b). 1D8$^{N5}$, 1D8$^{N18}$, and 1D8 IgG all bound to activated mouse CD8a$^+$ T cells but did not bind the unstimulated T cells (Fig. 2b). The binding of 1D8$^{N5}$ and 1D8$^{N18}$ was competitively inhibited by 1D8 IgG (Supplementary Figure 2c). These results show that the 1D8$^N$ trimerbodies retain the ability to bind to endogenous murine 4-1BB.

We proceeded to investigate the costimulatory capability of the 1D8$^N$ trimerbodies by testing their effect on the proliferation, IFN-γ secretion, and viability of mouse CD8a$^+$ T cells in the presence of a suboptimal dose of anti-CD3 mAb. 1D8 IgG and 1D8$^{N18}$ increased proliferation ($P = 0.0163$ and $P = 0.0013$, respectively) and IFN-γ secretion ($P = 0.0092$ and $P = 0.0101$, respectively) similarly to each other (Fig. 2c, d), and 1D8$^{N18}$ was significantly more potent than 1D8$^{N5}$ and 1D8$^{N0}$ ($P = 0.0234$ and $P = 0.0016$, respectively; Fig. 2c). After 72 h, a statistically significant increased viability of CD8a$^+$ T cells stimulated with 1D8 IgG and 1D8$^{N18}$ was observed ($P = 0.005$ and $P = 0.015$, respectively; Fig. 2e). Furthermore, 1D8$^{N18}$ was significantly more potent than 1D8 IgG ($P = 0.046$; Fig. 2e). The recombinant soluble mouse 4-1BBL (m4-1BBL) was essentially inactive (Fig. 2c, e). The m4-1BBL migrates at approximately 40 kDa in reducing conditions and at approximately 70 kDa in non-reducing conditions, compatible with a trimer (Supplementary Figure 3a). The binding to m4-1BB expressed on the cell surface was less efficient than the 1D8$^{N18}$ (Supplementary Figure 3b), and competition ELISA demonstrated that m4-1BBL and 1D8$^{N18}$ recognize different regions of the m4-1BB (Supplementary Figure 3c). Next, we investigated the spatio-temporal distribution and dynamics of the interactions between cell surface m4-1BB and CF488A-labeled m4-1BBL, 1D8 IgG or 1D8$^{N18}$ (Supplementary Figure 4) in living HEK293$^{m4-1BB}$-S cells displaying homogenous expression of the receptor (Supplementary

Figure 5). Employing raster imaging correlation spectroscopy (RICS)[34], we observed and quantified receptor clustering and its molecular mobility upon binding. We observed that m4-1BBL does not induce receptor clustering but rather internalization of the receptor into the cytoplasm ($\approx 70$ μm$^2$/s) (Supplementary Figure 6a). In contrast, both 1D8 IgG and 1D8$^{N18}$ induce cluster formation, reducing the lateral mobility drastically at the plasma membrane upon binding, from $\approx 1.5$ to 0.35 and from $\approx 1.0$ to 0.15 μm$^2$/s, respectively (Fig. 2f and Supplementary Figure 6b, c). 1D8$^{N18}$ formed larger and more numerous membrane clusters, which consequently impeded the lateral diffusion to a greater degree, which indicates more effective and extensive crosslinking (Fig. 2g).

As the 1D8$^{N18}$ demonstrated improved T cell costimulatory activity and induced receptor clustering to a greater degree, we chose this particular configuration for subsequent studies. We used size exclusion chromatography with multi-angle light scattering (SEC-MALS) to investigate the oligomeric state of 1D8$^{N18}$. It eluted as a major symmetric peak with a mass of 112 kDa (Supplementary Figure 7a), close to the predicted 110.1 kDa of 1D8$^{N18}$ without signal sequence (mass spectrometry by matrix assisted laser desorption/ionization (MALDI) confirmed its absence). Two minor peaks were also detected, the smallest being a protein impurity, as seen by sodium dodecyl sulfate-polyacrylamide gel electrophoresis (SDS-PAGE; Supplementary Figure 7b). The other peak migrates in SDS-PAGE as 1D8$^{N18}$ but has a native mass of 244 kDa, probably corresponding to trimer–dimers. The two species can be separated by SEC, but reinjection of the major peak gives another minor trimer–dimer peak (Supplementary Figure 7c), indicating an equilibrium where the trimeric species are predominant (85% and 97% at 1.0 and 0.26 g/l, respectively). The circular dichroism spectrum of 1D8$^{N18}$ (Supplementary Figure 7d) is typical of proteins with predominantly β-sheet structure. The 1D8$^{N18}$ is folded into a stable three-dimensional structure, as shown by the cooperative thermal denaturation ($T_m \approx 57$ °C; Supplementary Figure 7e). To understand the behavior and structure of 1D8$^{N18}$ in solution, we analyzed it in the absence of substrate using small-angle X-ray scattering (SAXS) (Fig. 2h and Supplementary Figure 7f, g, Supplementary Table 2). The ab initio model shows how the homotrimer adopts a pyramidal conformation stabilized by direct interactions between the C-terminal TIE$^{XVIII}$ domains, as has previously been described for human collagen$^{XVIII}$[35]. The 1D8 scFvs point orthogonally away from the plane defined by the trimerization domain without directly contacting each other, in a manner resembling an open tripod (Fig. 2h).

**Design of an EGFR-targeted 4-1BB-agonistic trimerbody.** We proceeded to generate a bispecific trimerbody by fusing the anti-human EGFR single-domain antibody ($V_{HH}$; EGa1)[32] to the C-terminus of 1D8$^{N18}$ through a 17-residue-long linker giving the 1D8$^{N/C}$EGa1 trimerbody (Fig. 1c). The construct was designed with a FLAG and a StrepII tag at the N-terminus of the 1D8 scFv. The 1D8$^{N/C}$EGa1 was produced in stably transfected HEK293 cells (5 mg/l), followed by Strep-Tactin affinity chromatography. SDS-PAGE analysis, under reducing conditions, of the purified protein revealed a single band with a molecular mass of 55.6 kDa consistent with the calculated from its amino acid sequence (52.9 kDa without the signal sequence; Supplementary Figure 8a). The oligomeric state of the purified 1D8$^{N/C}$EGa1 was examined by SEC-MALS measurements. The sample eluted as a major symmetric peak with a mass of 160 kDa, close to the calculated mass for a trimer without the signal sequence (158.7 kDa), and a minor peak with a mass of 309 kDa (Supplementary Figure 8b), which is indistinguishable from 1D8$^{N/C}$EGa1 by SDS-PAGE

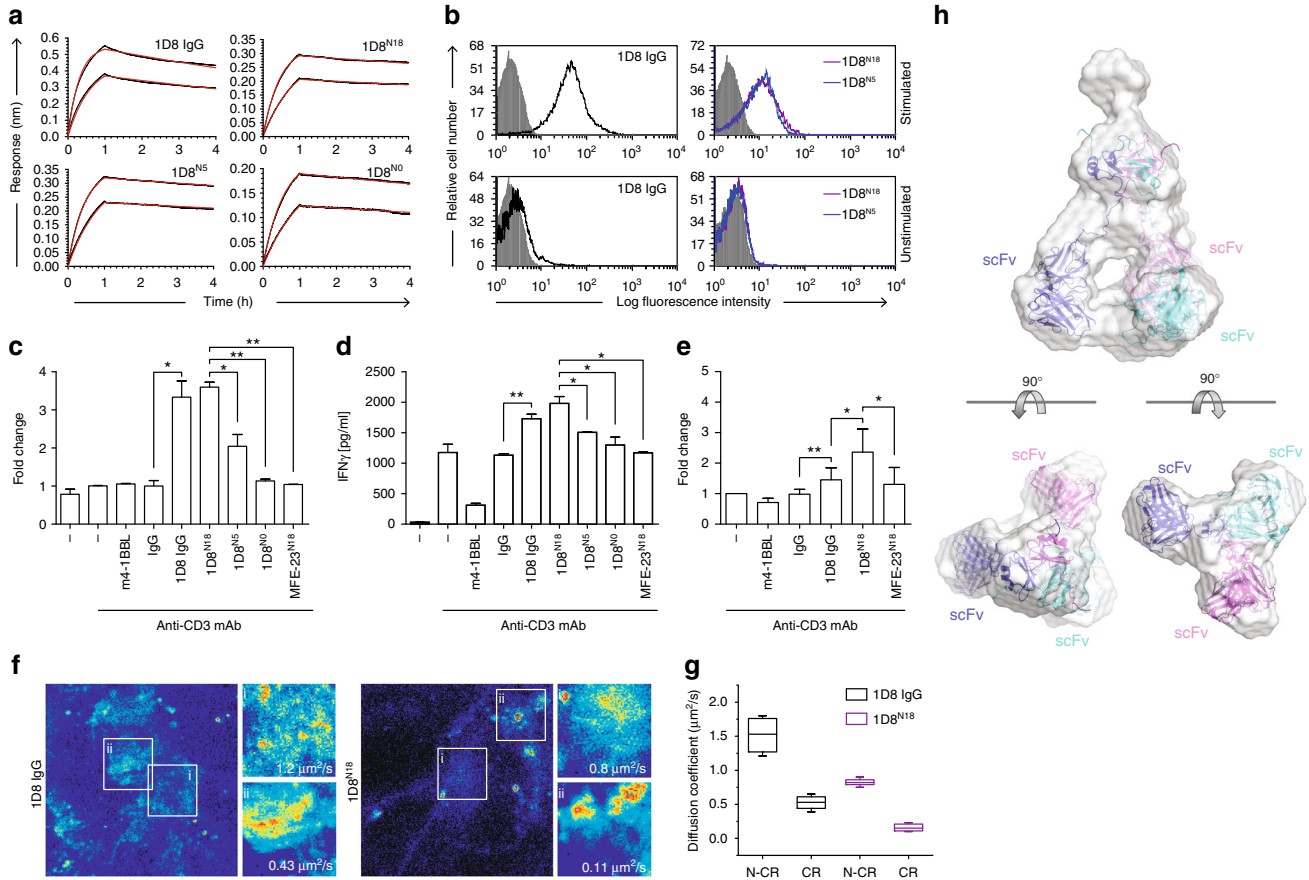

**Fig. 2** Characterization of anti-4-1BB trimerbodies. **a** Sensorgrams (black curves) and fitting curves for 1D8 antibodies (2 and 4 nM), obtained by biolayer interferometry with surface-immobilized m4-1BB. **b** The binding to 4-1BB on the cell surface of stimulated mouse CD8a$^+$ T cells measured by FACS. **c–e** Costimulatory activity of anti-4-1BB antibodies. Mouse CD8a$^+$ T cells were stimulated with immobilized anti-CD3 mAb in the presence of m4-1BBL, 1D8$^{N5}$, 1D8$^{N18}$, or 1D8 IgG, and proliferation (**c**) and secretion of IFN-γ (**d**) were measured after 48 h and cell viability (**e**) after 72 h. Data are reported as fold change relative to the values obtained from anti-CD3 mAb-stimulated CD8a$^+$ T cells. Rat IgG$_{2a}$ and MFE-23$^{N18}$ were used as controls. Data are mean ± SD ($n = 3$), *$P \leq 0.05$, **$P \leq 0.01$, Student's $t$ test. **f** RICS analyses performed in living HEK293$^{m4\text{-}1BB}$-S cells at regions containing clusters formed upon 1D8 IgG or 1D8$^{N18}$ addition and at regions where clusters where not present (insert and zoomed-in regions ii, and i, respectively). Representative maximum intensity projection maps showing the RICS analyzed regions of interest. Values in the zoomed-in regions show the diffusion coefficient of bound antibody. The color heat map indicates in blueish tones the lower intensity and in redder tones the higher intensity. **g** Statistical analysis of the quantified diffusion coefficient obtained from 5 to 7 independent live cell experiments and 3–5 different regions of interest per cell (N-CR non-clustered region, CR clustered region). Data are presented as median (center line), upper and lower quartiles (boxes), and minimum-maximum (whiskers). **h** Arrangement of 1D8$^{N18}$ trimerbody in solution, as determined by SAXS. Rigid-body overlaying of the ab initio-determined SAXS envelope for 1D8N$^{18}$. The generated model (where each chain is colored in blue, magenta, or cyan) fits into the envelope (colored in pale gray)

(Supplementary Figure 8c). These results again indicate the formation of a minor population of dimers of trimers, as seen for 1D8$^{N18}$. The two oligomeric species can be separated by SEC, and reinjection of the isolated major peak of trimers yields only a very minor peak at the elution volume of the trimer–dimers (Supplementary Figure 8d). 1D8$^{N/C}$EGa1 performed very similarly to 1D8$^{N18}$ in CD and cooperative thermal denaturation studies (Supplementary Figure 8e, f). SAXS showed that 1D8$^{N/C}$EGa1 contains the same trimerized TIE$^{XVIII}$ core seen for 1D8$^{N18}$. The binding domains, however, are distributed evenly around the plane defined by the TIE$^{XVIII}$ core (Fig. 3a and Supplementary Figure 8g, h, Supplementary Table 2), resembling a playground roundabout. Using the SAXS data and structural information from the protein data bank (PDB), we created a homology model intended to clarify the layout of the six binding domains around the core, as the linkers' length and flexibility permit several configurations, e.g., with the three 1D8 scFvs opposite the three EGa1 V$_{HH}$ (as shown in Fig. 3a), or with alternating 1D8 and EGa1 domains side-by-side each other. Unfortunately, the

obtained resolution is insufficient to distinguish between these possibilities, although the overall hexagonal structure is clearly defined in the ab initio SAXS model.

The functionality of the 1D8$^{N/C}$EGa1 was demonstrated by BLI. The 1D8$^{N/C}$EGa1 trimerbody has kinetic rate constants that are very similar to the 1D8$^{N}$ trimerbodies in its interaction with immobilized m4-1BB (Fig. 3b, Supplementary Table 1). The binding kinetics of 1D8$^{N/C}$EGa1 to immobilized human EGFR-Fc chimera (hEGFR) was also investigated by BLI, and the interaction was found to also have a low picomolar $K_D$ (Fig. 3b, Supplementary Table 1). Our previous comparison of EGa1 V$_{HH}$ and EGa1-derived N-trimerbody (EGa1$^{N}$) kinetics showed a low nanomolar $K_D$ for the EGa1 V$_{HH}$ and a low picomolar $K_D$ for EGa1$^{N}$ [36]. These kinetics are easily distinguishable, and 1D8$^{N/C}$EGa1 showed comparable kinetics to EGa1$^{N}$, indicating that it trivalently binds hEGFR. The 1D8$^{N/C}$EGa1 was found to be capable of binding immobilized m4-1BB and hEGFR simultaneously (Fig. 3c). This further demonstrates the bispecificity of 1D8$^{N/C}$EGa1 and shows a lack of steric hindrance between

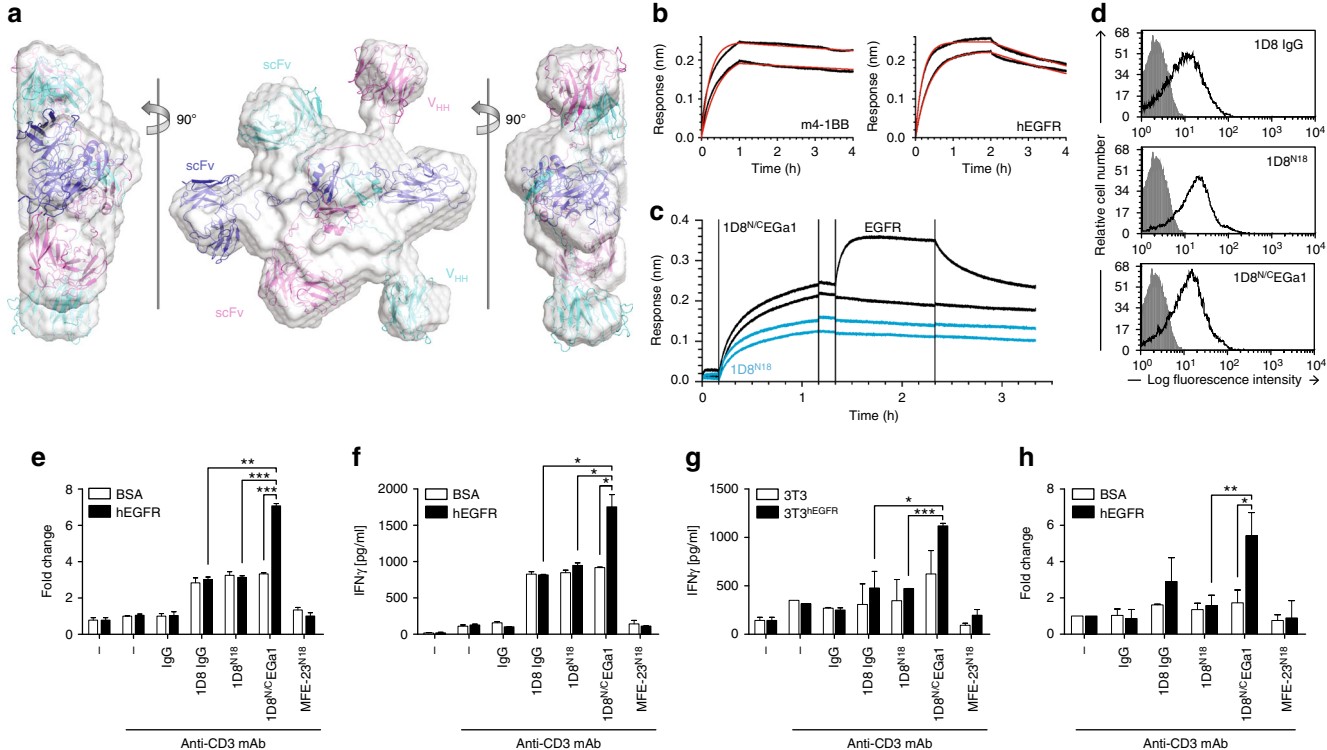

**Fig. 3** Characterization of the EGFR-targeted 4-1BB-agonistic trimerbody. **a** Arrangement of 1D8$^{N/C}$EGa1 trimerbody in solution by SAXS. Rigid-body fitting of the model corresponding to 1D8$^{NC}$EGa1 inside the SAXS envelope (colored in pale gray). Each chain has been colored in blue, magenta, or cyan. **b** Sensorgrams (black curves) and the results of fitting to a 1:1 model (red curves) obtained using biolayer interferometry for the interaction of 1D8$^{N/C}$EGa1 (2 and 4 nM) with immobilized m4-1BB and the interaction of 1D8$^{N/C}$EGa1 (0.5 and 1 nM) with immobilized hEGFR. **c** Simultaneous binding to both m4-1BB and hEGFR was demonstrated for 1D8$^{N/C}$EGa1 but not 1D8$^{N18}$. Biosensors were coated with m4-1BB, after which 4 nM of 1D8$^{N/C}$EGa1 (black curves) or 1D8$^{N18}$ (blue curves) were added. **d** The binding of anti-4-1BB antibodies to m4-1BB on the cell surface of stimulated mouse CD8a$^+$ T cells measured by FACS. **e**, –**f** Mouse CD8a$^+$ T cells were plated with immobilized anti-CD3 mAb and hEGFR or BSA in the presence of 1D8$^{N18}$, 1D8$^{N/C}$EGa1, or 1D8 IgG, and proliferation (**e**) and IFN-γ secretion (**f**) were determined after 48 h. EGFR-negative 3T3 cells or EGFR-positive 3T3$^{hEGFR}$ cells were co-cultured with mouse CD8a$^+$ T cells in the presence of anti-CD3 mAb and 1D8$^{N18}$, 1D8$^{N/C}$EGa1, or 1D8 IgG. IFN-γ secretion analyzed after 48 h (**g**) and cell viability after 72 h (**h**). Data are represented as fold change relative to the values obtained from anti-CD3 mAb stimulated cells. Rat IgG$_{2a}$ and MFE-23$^{N18}$ were used as negative controls. Data are mean ± SD ($n = 3$), *$P ≤ 0.05$, **$P ≤ 0.01$, ***$P ≤ 0.001$, Student's $t$ test

its interactions with hEGFR and m-4-1BB. The ability of 1D8$^{N/C}$EGa1 to detect its antigens as cell surface proteins was studied by flow cytometry. The 1D8$^{N/C}$EGa1 trimerbody bound to HEK293 (EGFR+), to HEK293$^{m4-1BB}$ cells, and to mouse 3T3 cells expressing human EGFR (3T3$^{hEGFR}$) but not to wild-type 3T3 cells (Supplementary Figure 9). Furthermore, 1D8$^{N/C}$EGa1 bound to activated mouse CD8a$^+$ T cells as efficiently as the 1D8$^{N18}$ (Fig. 3d). To further assess the multivalent binding of 1D8$^{N/C}$EGa1, we studied its capacity to inhibit proliferation and EGFR phosphorylation in A431 cells[32]. 1D8$^{N/C}$EGa1 and cetuximab, an EGF-competitive inhibitor[37], but neither the anti-human CD20 rituximab nor 1D8 IgG, inhibited A431 proliferation, in a dose-dependent manner ($P ≤ 0.0001$ for the higher doses of both antibodies, vs. equimolar doses of control antibodies; Supplementary Figure 10a), and EGFR phosphorylation (Supplementary Figure 10b).

We then wanted to determine whether 1D8$^{N/C}$EGa1 retained the baseline costimulatory capacity seen for 1D8$^{N18}$ and whether this was improved by the crosslinking provided through EGa1's binding to hEGFR. CD8a$^+$ T cells were stimulated with immobilized anti-CD3 mAb and the panel of costimulatory agents in solution, in the presence or absence of plastic-immobilized hEGFR. The 1D8$^{N/C}$EGa1 had a costimulatory effect similar to 1D8$^{N18}$ in the absence of hEGFR, but proliferation ($P = 0.0008$) and IFN-γ levels ($P = 0.0198$) were greatly enhanced when hEGFR was included (Fig. 3e, f). This

effect was further confirmed by co-culture assays using EGFR-negative and EGFR-positive target cells. The IFN-γ levels were significantly higher when CD8a$^+$ T cells were co-cultured with 3T3$^{hEGFR}$ in the presence of the 1D8$^{N/C}$EGa1, as compared to the non-targeted 1D8 molecules ($P = 0.0344$ 1D8 IgG, and $P = 0.0009$ 1D8$^{N18}$; Fig. 3g). We then investigated the effect of EGFR-targeted 4-1BB costimulation on cell viability. After 72 h, a statistically significant increased viability of CD8a$^+$ T cells costimulated with 1D8$^{N/C}$EGa1 in the presence of plastic-immobilized hEGFR was observed ($P = 0.0326$), as compared to cells costimulated with 1D8$^{N18}$ ($P = 0.0088$; Fig. 3h).

**The EGFR-targeted trimerbody shows high tumor localization.** First, we studied the serum stability of 1D8$^{N18}$ and 1D8$^{N/C}$EGa1, and no significant loss of 4-1BB- or EGFR-binding activity was detected even after 7 days in mouse serum at 37 °C (Supplementary Figure 11a, b). Pharmacokinetic studies were performed in immunocompetent mice, which received a single intravenously (i.v.) injection of the anti-4-1BB antibodies. The serum concentrations of active protein were determined by ELISA with immobilized m4-1BB. In CD-1 mice, the 1D8$^{N18}$ was rapidly cleared from circulation with a terminal half-life of 1.3 h, whereas the 1D8$^{N/C}$EGa1 showed a prolonged circulatory half-life of 16.1 h (Fig. 4a, Supplementary Table 3). 1D8$^{N/C}$EGa1 serum half-life was not influenced by the genetic background of the mice, and

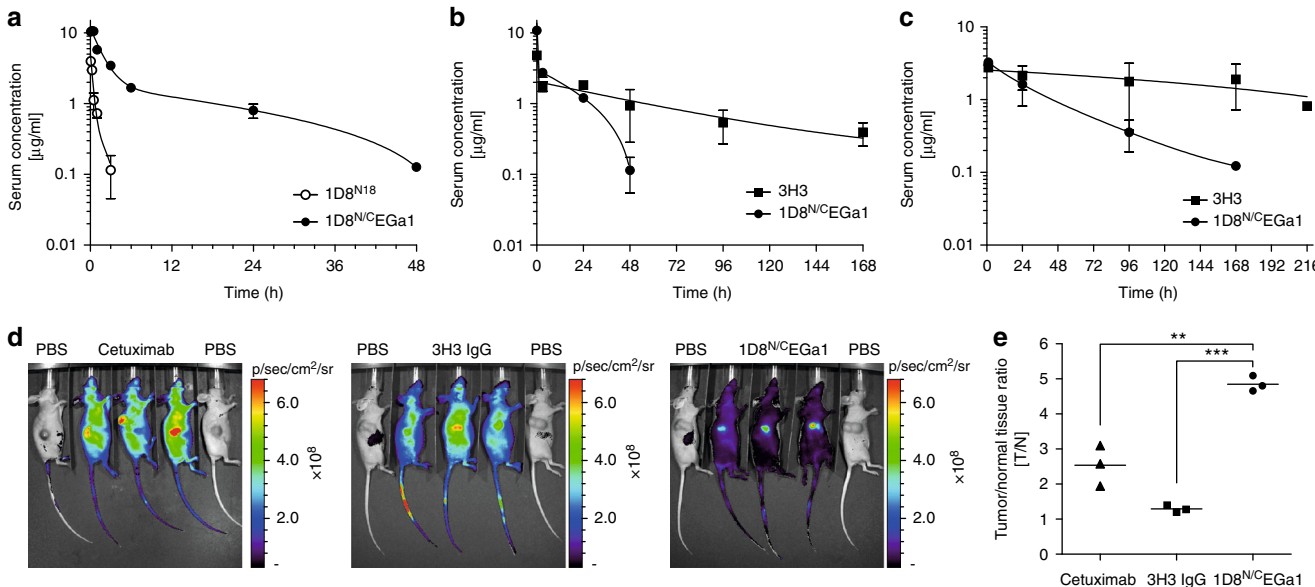

**Fig. 4** Pharmacokinetic properties and tumor imaging of the EGFR-targeted 4-1BB-agonistic trimerbody. Pharmacokinetic studies after a single i.v. dose (1 mg/Kg) of 1D8[N18] or 1D8[N/C]EGa1 in CD-1mice (**a**) or of 3H3 IgG or 1D8[N/C]EGa1 in BALB/c mice (**b**). Pharmacokinetic study after a single i.p. dose (1 mg/Kg) of 3H3 IgG or 1D8[N/C]EGa1 in BALB/c mice (**c**). **d** In vivo fluorescence imaging of A431 tumor-bearing nude mice 24 h after i.v. injection of PBS or 100 µg of Cy5-labeled cetuximab, CF647-labeled 3H3 IgG or CF647-labeled 1D8[N/C]EGa1. **e** Tumor to normal tissue (T/N) ratios. The color scale bar represents the fluorescence intensity recorded as photons per second per centimeter squared per steradian (p/s/cm²/sr). Data are mean ± SD ($n = 3$), **$P \leq 0.01$, ***$P \leq 0.001$, Student's $t$ test

a similar pharmacokinetic profile was observed in BALB/c mice (Fig. 4b, Supplementary Table 4). The half-life of the anti-4-1BB mAb 3H3[33] was 4.8 days, consistent with published data[38] (Fig. 4b, Supplementary Table 4). After a single intraperitoneal (i.p.) injection in BALB/c mice, the half-lives of 1D8[N/C]EGa1 and 3H3 IgG were 32 h and 7 days, respectively (Fig. 4c, Supplementary Table 5). For in vivo imaging, 3H3 IgG, 1D8[N/C]EGa1, and the anti-EGFR mAb cetuximab were labeled with near-infrared (NIR) fluorochromes, which did not change their SDS-PAGE migration or compromise their binding to cell surface antigen (Supplementary Figure 12). Athymic nude mice bearing hEGFR-positive A431 tumor xenografts subcutaneously (s.c.) implanted into the right flank were i.v. injected in the tail vein with NIR-labeled antibodies and imaged 24 h later (Fig. 4d). The 1D8[N/C]EGa1 trimerbody showed high tumor localization with a tumor to normal tissue (T/N) ratio of 4.85 ± 0.13 (mean ± SD), as compared to that of cetuximab (2.54 ± 0.34) ($P \leq 0.01$) and 3H3 IgG (1.29 ± 0.06) ($P \leq 0.0001$), which corresponds to little to no specific tumor accumulation (Fig. 4d, e).

**Antitumor activity of EGFR-targeted 4-1BB-agonistic trimerbody**. To study the antitumor effects of the EGFR-targeted 4-1BB-agonistic trimerbody in immune competent mice, we used murine CT26 colorectal carcinoma (H-2d) cells infected with retrovirus encoding human EGFR (CT26[hEGFR]) (Supplementary Figure 13a). The in vitro cell proliferation rates and the in vivo take rate and growth curves in BALB/c mice of CT26[hEGFR] cells and CT26[mock] cells infected with empty vector retroviruses were similar (Supplementary Figure 13b, c), suggesting that the expression of hEGFR did not significantly alter the poor immunogenicity of the CT26 tumor cells. Furthermore, ex vivo isolated CT26[hEGFR] cells from 3-week-old s.c. tumors expressed significant levels of surface hEGFR (Supplementary Figure 13d). To elucidate the functionality of the EGFR pathway in CT26[hEGFR] cells, we studied the capacity of cetuximab and 1D8[N/C]EGa1 to inhibit their proliferation. As shown in Supplementary Figure

13e, neither cetuximab nor 1D8[N/C]EGa1 had a significant effect on CT26[hEGFR] proliferation ($P = 0.6647$ and $P = 0.0760$ respectively, for higher dose, vs. equimolar dose of control antibody). Therefore, the potential therapeutic effect of 1D8[N/C]EGa1 is not contributed by an EGa1-mediated antiproliferative effect. We used an established regimen to administer IgG-based 4-1BB-agonistic mAbs and the EGFR-targeted 4-1BB-agonistic trimerbody, with three i.p. injections at 2-day intervals[15]. Injection of purified 1D8[N/C]EGa1 in mice bearing established CT26[hEGFR] tumors (average diameter of 0.4 cm) induced tumor regression in 8 out of 10 (80%) mice in two separate experiments (Fig. 5a, b). Treatment with the IgG-based 4-1BB agonist antibodies 1D8 (Fig. 5a) or 3H3 (Fig. 5b) resulted in complete regression in 10 out of 11 (91%) mice bearing CT26[hEGFR] tumors. All mice treated with phosphate-buffered saline (PBS), control antibodies (isotype control rat IgG_{2a} and MFE-23[N18]), and 1D8[N18] were sacrificed within 4–5 weeks after tumor cell implantation (Fig. 5a, b). It is well established that mice cured by IgG-based 4-1BB-agonistic mAb treatment have long-lasting and tumor-specific immunity[39–41]. To investigate whether the EGFR-targeted 4-1BB-agonistic trimerbody can generate a similar effect, mice that rejected the implanted CT26[hEGFR] tumor by treatment with 3H3 IgG or 1D8[N/C]EGa1 (Fig. 5b) were rechallenged 50 days later with CT26[mock] cells. 3H3 IgG- and 1D8[N/C]EGa1-cured mice ($P = 0.0027$ and $P = 0.0067$, respectively), but not age-matched naive mice, were resistant to a rechallenge with CT26[mock] tumor cells (Fig. 5c and Supplementary Figure 14), showing that the trimerbody-mediated EGFR-targeted 4-1BB costimulation can induce long-term protective immunological memory against CT26 tumors that do not express hEGFR. In order to understand the antitumor immune response generated with 4-1BB antibodies, tumors from 3H3 IgG- and 1D8[N/C]EGa1-treated mice and control mice were extracted 2 days after receiving the third i.p dose (day 13 after tumor inoculation) (Supplementary Figure 15) for immunohistochemistry to quantify tumor-infiltrating CD8 + T lymphocytes (TILs). The percentage of CD8+ TILs was an order of magnitude higher in 3H3 IgG- and

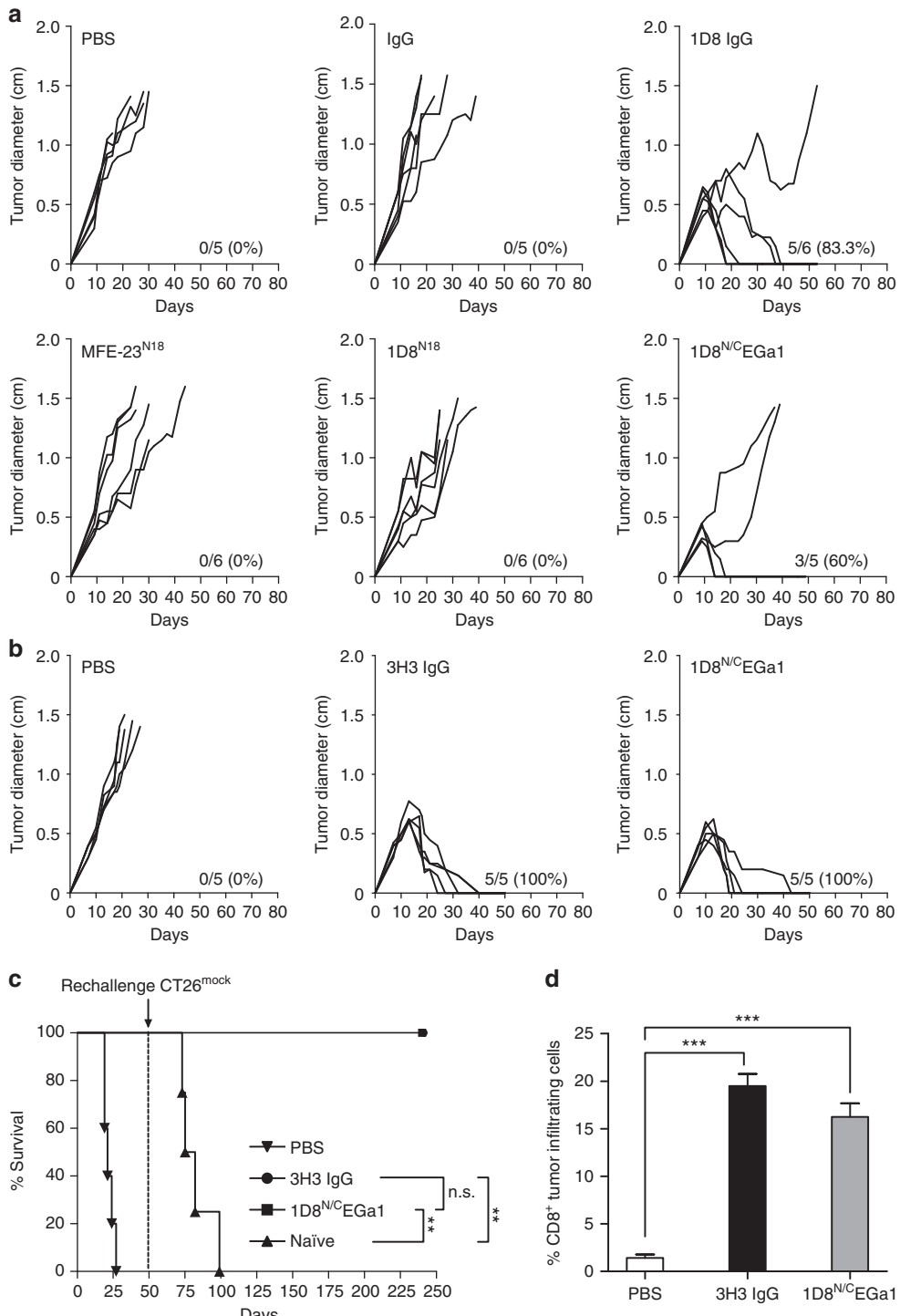

**Fig. 5** Induction of tumor regression in mice treated with 1D8$^{N/C}$EGa1 trimerbody. **a** BALB/c mice inoculated s.c. with CT26$^{hEGFR}$ tumor cells ($n = 6$/group) were treated with three i.p. doses (4 mg/kg) of rat IgG$_{2a}$ isotype, 1D8 IgG, MFE-23$^{N18}$, 1D8$^{N18}$, 1D8$^{N/C}$EGa1, or with PBS and monitored for tumor growth. Tumor diameter growth curves for individual mice in each treatment group are presented. The results are representative of two experiments identically performed. **b** BALB/c mice bearing CT26$^{hEGFR}$ tumors ($n = 5$/group) were treated with three i.p. doses of either PBS, 3H3 IgG, or 1D8$^{N/C}$EGa1. **c** Kaplan–Meier survival curves of the 1D8$^{N/C}$EGa1 trimerbody-treated mice (**$P \le 0.01$), log-rank (Mantel–Cox) test. Long-term survivors, following complete tumor rejection (**b**) were rechallenged with CT26$^{mock}$ cells (s.c.) 50 days after i.p. injections of 3H3 IgG or 1D8$^{N/C}$EGa1 trimerbody. As a control group, tumor naive mice developed tumors in every case. **d** Quantitative analysis of intratumoral CD8$^+$ T cells in paraffin-embedded mouse tumor tissue ($n = 3$/group) by immunohistochemistry. Data were calculated as percentage of CD8$^+$ versus total cell number and presented as mean ± SD. ***$P \le$ 0.001, Student's $t$ test

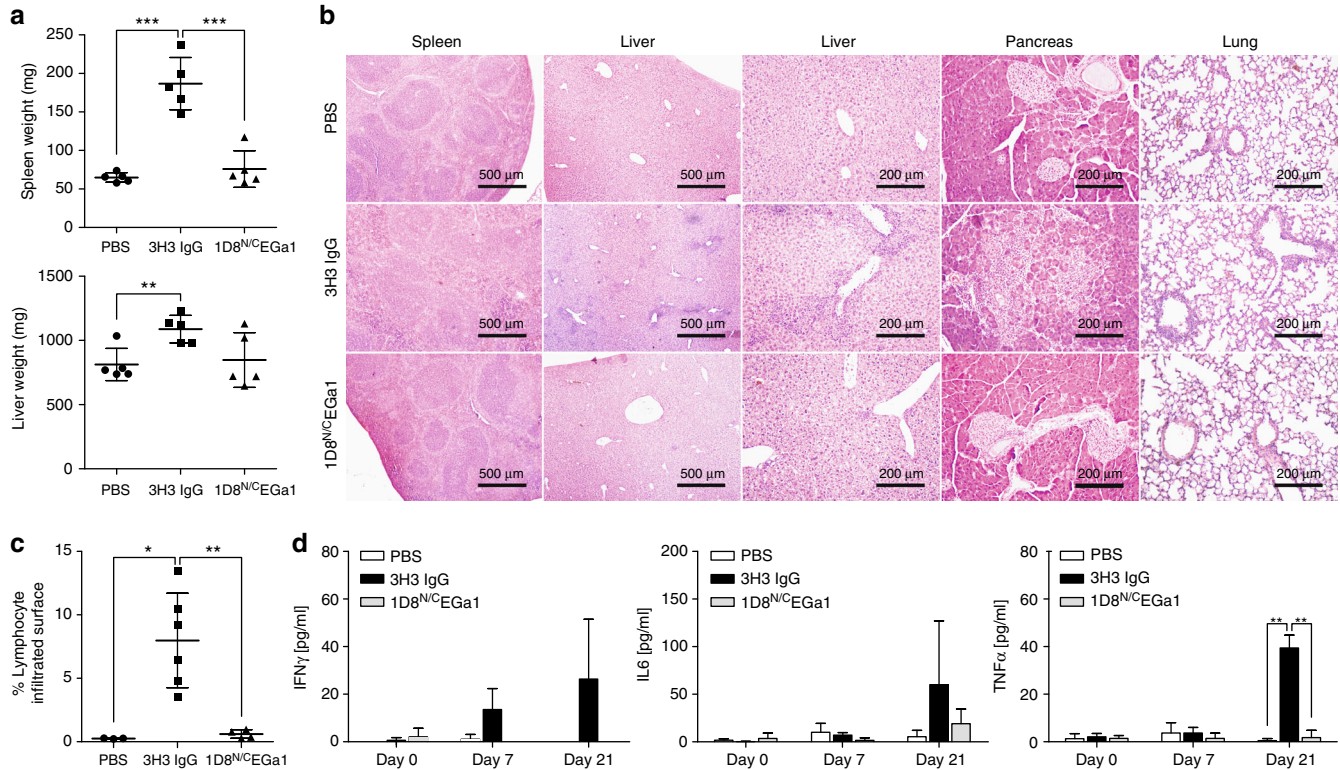

**Fig. 6** Treatment with 1D8$^{N/C}$EGa1 does not induce toxicity. **a** Spleens and liver weights from mice ($n = 5$/group) treated with PBS, 3H3 IgG, or 1D8$^{N/C}$EGa1. **b** Hematoxylin and eosin staining of representative tissue slides from the spleen, liver, pancreas, and lung of mice treated with PBS, 3H3 IgG, and 1D8$^{N/C}$EGa1. Magnification is ×40 (spleen and liver) and ×200 (liver, pancreas, and lung). Scale bars are shown. **c** Quantification of the mononuclear cell-infiltrated surface in the liver of mice treated with PBS ($n = 3$), 3H3 IgG ($n = 5$), or 1D8$^{N/C}$EGa1 ($n = 4$). **d** Sera from treated mice were collected from peripheral blood at days 0, 7, and 21 of treatment, and levels of INF-γ, TNF-α, and IL-6 were measured by luminex assays ($n = 3$ per time point). All data are presented as mean ± SD. $P$ values (*$P \leq 0.05$, **$P \leq 0.01$, ***$P \leq 0.001$) were calculated with Student's $t$ test

1D8$^{N/C}$EGa1-treated mice (19.48 ± 1.28 and 16.25 ± 1.41, respectively) compared to the PBS-treated mice (1.42 ± 0.36) (Fig. 5d and Supplementary Figure 16), indicating an efficient tumor recruitment of CD8$^+$ T lymphocytes in the antibody-treated mice.

**EGFR-targeted 4-1BB-agonistic trimerbody shows negligible toxicity.** We compared the toxicity profile of 1D8$^{N/C}$EGa1 directly with 3H3 IgG under similar conditions. Mice were injected i.p. with PBS, 3H3 IgG, or 1D8$^{N/C}$EGa1 (6 mg/kg) once a week for 3 weeks and euthanized 1 week later. As shown in Fig. 6a, treatment with 3H3 IgG resulted in gross enlargement of spleen and livers as demonstrated by weight ($P \leq 0.0001$ and $P = 0.0058$, respectively). In contrast, treatment with 1D8$^{N/C}$EGa1 did not result in splenomegaly or hepatomegaly (Fig. 6a). The histologic study of the spleens of mice treated with 3H3 IgG revealed an altered architecture with expanded follicles with undefined zones and clear evidence of extramedullary hematopoiesis (Fig. 6b), as previously described[14]. In contrast, the spleens of mice treated with 1D8$^{N/C}$EGa1 showed normal histology (Fig. 6b). Also confirming previous results[14,15], treatment with 3H3 IgG caused significant mononuclear cell infiltration in the liver, forming periportal cuffs with thickening of tunica media and also infiltration foci associated with microvasculature, while no significant infiltration was observed in mice treated with 1D8$^{N/C}$EGa1 (Fig. 6b). Indeed, the surface of infiltrating mononuclear cells accounted for 8% of the liver of mice treated with 3H3 IgG, while it only represented 0.6% in mice treated with 1D8$^{N/C}$EGa1 ($P = 0.0048$) and 0.25% in mice treated with PBS

($P = 0.0104$) (Fig. 6c). In addition, staining of collagen fibers with sirius red (Supplementary Figure 17) or Masson's trichrome (Supplementary Figure 18) showed that 3H3 IgG treatment, but not 1D8$^{N/C}$EGa1, caused the deposition and disarrayment of portal collagen fibers, indicative of an early stage of fibrosis. Mononuclear cell infiltration was also seen in the lungs, forming perivascular cuffs, and in pancreas of 3H3 IgG-treated mice. Pancreas infiltration radiated from the vasculature and extended to the neighboring intercalated ducts (Fig. 6b). Prominent collagen deposition in the infiltrated areas of the pancreas of mice treated with 3H3 IgG was observed, indicative of fibrosis. In contrast, none of these features were observed in mice treated with 1D8$^{N/C}$EGa1 (Supplementary Figure 17 and 18). The effects of treatment with 3H3 IgG and 1D8$^{N/C}$EGa1 on the levels of pro-inflammatory cytokines in serum were also compared. As shown in Fig. 6d, 3H3 IgG treatment triggered significant elevation of INF-γ, IL-6, and TNF-α ($P = 0.0055$ PBS, and $P = 0.0015$ 1D8$^{N/C}$EGa1), particularly evident at day 21. In contrast, 1D8$^{N/C}$EGa1 induced minimal or undetectable levels of inflammatory cytokines comparable to PBS-treated animals.

In order to investigate whether 1D8$^{N/C}$EGa1's shorter half-life might be responsible for its lack of toxicity, another study was conducted in BALB/c mice in which 1D8$^{N/C}$EGa was administered (6 mg/kg) i.p. every 3 or 4 days, for 3 weeks, for a total of six doses. These more frequent injections were intended to maintain circulatory levels of 1D8$^{N/C}$EGa1 comparable to those of the longer-circulating 3H3 IgG, which was injected as before with 6 mg/kg i.p injections once weekly for 3 weeks. The escalated 1D8$^{N/C}$EGa1 regimen did not induce splenomegaly or hepatomegaly, nor significant histological alterations, while the standard 3H3

IgG regimen induced alterations similar to those observed in 3H3 IgG-treated C57BL/6 mice (Supplementary Figure 19).

## Discussion

In this study, we describe a tumor-targeted 4-1BB agonist trimerbody with similar efficacy and less toxicity than conventional IgG-based 4-1BB-agonistic antibodies. This hexavalent trimerbody consists of three anti-4-1BB scFvs and three anti-EGFR single-domain antibodies organized around a modified homo-trimerization domain from collagen XVIII. More than 80% of its mass is directly involved in antigen binding. While the mono-specific trivalent 1D8[N18] trimerbody adopts a tripod-shaped conformation, the addition of the anti-EGFR $V_{HH}$ antibodies into the bispecific hexavalent 1D8[N/C]EGa1 trimerbody changes the conformation to an extended and planar hexagram-shaped structure with the six binding domains exposed at the periphery. This structure is inherently flexible, and the SAXS ab initio model supports a dynamic equilibrium of the open conformation where the binding domains are not locked into place relative to the core but instead exist in various extended conformations. These structural findings, when considered alongside the high avidity observed in characterization of 1D8[N/C]EGa1's binding to biosensor- and cell surface-displayed 4-1BB, indicate that the binding domains are predominantly sterically unhindered and available for antigen binding.

The 1D8[N/C]EGa1 trimerbody is efficiently expressed by transfected human cells as soluble protein and can be purified using standard affinity chromatographic methods. SEC-MALS showed that both 1D8[N18] and 1D8[N/C]EGa1 trimerbodies primarily formed the expected trimeric structure, with a minor fraction forming higher-order oligomers (likely trimer–dimers). This effect was not observed with either EGa1 $V_{HH}$-based trimerbodies[36,42] or non-1D8 scFv-based trimerbodies[31,43] and is therefore likely attributable to the 1D8 scFv. The partial dimerization of scFvs[44] and clustering behavior of certain scFvs in particular[45] have previously been reported, so this is not a new phenomenon.

The binding experiments provide quantitative evidence for trimeric interactions between the 1D8[N/C]EGa1 trimerbody and both EGFR and 4-1BB. This 3:3 stoichiometry of the 1D8[N/C]EGa1 trimerbody is unique among the existing body of 4-1BB-binding proteins, including 4-1BBL fusion proteins[25]. We have provided evidence that, upon binding of soluble 4-1BBL to cell surface-displayed 4-1BB, the complex is rapidly internalized, whereas the anti-4-1BB antibodies induced the formation of durable clusters. The amount and stability of the 4-1BB clusters are significantly higher upon interaction with trivalent 1D8[N18] trimerbody than with bivalent 1D8 IgG, and this, along with the increased avidity for EGFR, could explain the increased costimulation induced by 1D8[N/C]EGa1. The bispecific trimerbody demonstrated better costimulation of CD8a$^+$ T cells in the presence of human EGFR, either immobilized or expressed in a cell surface context. These results support the 1D8[N/C]EGa1 trimerbody-induced formation of dense, clustered 4-1BB signaling complexes at the point of contact between a T cell and a TAA-displaying surface. The need for 4-1BB crosslinking beyond trimerization (i.e., hyper-crosslinking) has previously been reported as necessary for inducing strong 4-1BB signaling[20].

Besides the potent tumor-specific costimulation, the 1D8[N/C]EGa1 trimerbody exhibit improved serum stability as well as efficient and rapid localization to EGFR-positive tumors. The 1D8[N/C]EGa1 trimerbody may be able to maximize tumor-specific costimulation due to the combination of a higher total tumor uptake and substantially faster circulatory clearance, ultimately giving an improved T/N ratio as compared to conventional IgG-

based anti-EGFR and anti-4-1BB antibodies. The magnitude of the protective antitumor response provided by the EGFR-targeted 4-1BB-agonistic trimerbody is equivalent to that observed for two well-characterized anti-4-1BB mAbs, 1D8 and 3H3[33]. We show that the administration of the 1D8[N/C]EGa1 trimerbody can eradicate established CT26[hEGFR] tumors. Furthermore, rechallenging of the mice that rejected the tumor upon treatment with EGFR-targeted 4-1BB-agonistic trimerbody with fresh CT26[mock] cells failed to allow tumor implantation, thus indicating that long-lasting tumor-specific memory had been established. In this study, the 4-1BB agonists were administered intraperitoneally, using a treatment regimen that has been validated for IgG-based anti-4-1BB antibodies[15]. Despite its significantly shorter terminal half-life, the 1D8[N/C]EGa1 trimerbody showed similar efficacy to 1D8 and 3H3 mAbs, indicating that its quicker clearance is compensated for by other factors, potentially improved tumor retention due to higher avidity, and more extensive tumor penetration due to its structure. As 1D8[N/C]EGa1 did not induce toxicity even when circulating at high concentrations for a 3-week period, half-life extension strategies can be explored, such as inclusion of albumin fragments or albumin-binding moieties[46]. The mechanism of action of IgG-based 4-1BB-agonistic antibodies has been interpreted as a consequence of enhanced anti-tumor CD8$^+$ T cell responses[47], and as the percentage of infiltrating CD8$^+$ T cells is similar in 3H3 IgG- and 1D8[N/C]EGa1-treated mice, it is likely that both IgG-based anti-4-1BB antibodies and tumor-targeted 4-1BB agonist trimerbody act in a mechanistically equivalent manner.

Toxicity has been the major impediment for first-generation 4-1BB-agonistic mAbs. In preclinical murine models, systemic administration of IgG-based anti-4-1BB antibodies resulted in nonspecific immune stimulation and other immune-related anomalies that affected the function of organs such as the liver, spleen, and bone marrow[14,48]. Similarly to these mouse studies, clinical trials using urelumab have been associated with adverse effects at higher doses, such as liver toxicity that resulted in two fatalities[16]. Here we show that treatment of naive immuno-competent mice with the IgG-based anti-4-1BB-agonistic antibody resulted in severe toxicity, as assessed by enlarged spleen and liver, severe inflammation and fibrosis in the liver, spleen and pancreas, and systemic inflammatory cytokine production. Treatment with an EGFR-targeted 4-1BB-agonistic trimerbody lacked these immune-related side effects, even after an escalated regimen doubling the number of injections. Our study suggests that these immunological abnormalities and organ toxicities are mainly dependent on FcR interactions.

In summary, bispecific trimerbodies have been effectively applied to the treatment of established tumors by tumor-targeted 4-1BB costimulation and comprise a novel technology platform for the future development of this and similar immunotherapeutic approaches, e.g., tumor-specific costimulation through CD40 or OX40, and combining PD-L1 blocking with costimulatory strategies. The EGFR-targeted 4-1BB agonist trimerbody described in this study was successfully implemented from its design, as it had the desired structure, function, and stability. In vivo, it did not show the toxicities that have been associated with IgG-based 4-1BB agonist and have notoriously held them back in clinical trials. Our results emphasize the need for future studies with non-canonical immunotherapeutic antibodies in order to realize the therapeutic potential of 4-1BB costimulation.

## Methods

**Mice.** C57BL/6, BALB/c, Hsd:ICR (CD-1), and Hsd:athymic Nude-Foxn1nu female mice were purchased from Harlan Iberica. Animals were housed in controlled conditions of temperature (21 ± 1 °C), humidity (50 ± 5%), and 12 h light/dark cycles. Manipulation was performed in laminar flow hood, when necessary,

and sterilized water and food were available ad libitum. All animal procedures conformed to European Union Directive 86/609/EEC and Recommendation 2007/526/EC, enforced in Spanish law under RD 1201/2005. Animal protocols were approved by the respective Ethics Committee of Animal Experimentation of the participant institutions: Instituto Investigación Sanitaria Puerta de Hierro-Segovia de Arana (Hospital Universitario Puerta de Hierro Majadahonda, Madrid, Spain), Intituto de Investigaciones Biomédicas "Alberto Sols" (IIBm) (CSIC-UAM, Madrid), and Laboratory Animal Applied Research Platform (Parc Científic, Barcelona, Spain). Procedures were additionally approved by the Animal Welfare Division of the Environmental Affairs Council of the Government of Madrid (66/14, 430/15, 264/16) and by the Ethics Committee of the Catalonian authorities (9912).

**Cells and culture conditions**. HEK293 (CRL-1573), NIH/3T3 (CRL-1658), and A431 (CRL-1555) cells were obtained from ATCC and were grown in complete Dulbecco's modified Eagle's medium (DMEM) at 37 °C. HEK293 cells expressing m4-1BB (HEK239[m4-1BB])[49]were cultured in complete DMEM+500 μg/ml G418. NIH/3T3 cells expressing human EGFR (3T3[hEGFR]) were provided by Dr A. Villalobo (IIBm, Madrid)[50]. Mouse CT26 cells (CRL-2638) infected with p-BABE-puro-hEGFR[51]expressing human EGFR (CT26[hEGFR]) or infected with the empty vector retrovirus (CT26[mock]) were provided by Dr M. Rescigno (European Institute of Oncology, Milan)[37]. The cell lines were routinely screened for mycoplasma contamination by PCR (Stratagene).

**Construction of expression vectors**. A synthetic gene encoding the 1D8 scFv was synthesized by Geneart AG and subcloned as ClaI/NotI into pCR3.1-L36-NC[31]resulting in pCR3.1-1D8-TIE[18]. To obtain the plasmids pCR3.1-1D8-TIE[0] and pCR3.1-1D8-TIE[5], two synthetic genes encoding the 1D8 scFv gene fused directly or by a 5-mer flexible linker to the N-terminus of the mouse TIE[XVIII] domain were synthesized by Geneart AG and subcloned as PstI/XbaI into pCR3.1-1D8-TIE[18]. To generate the bispecific trimerbody-expressing vector, the BamHI/XbaI fragment, containing the EGa1 gene, from pCR3.1-EGa1-TIE[7][44] was ligated into pCR3.1-1D8-TIE[18] to obtain pCR3.1-1D8[N18]-TIE-[C18]EGa1-myc/His. In order to introduce a N-terminal FLAG-StrepII tag, the HindIII/NotI fragment from pCR3.1-FLAG-StrepII-1D8-TIE[18]-iRGD was ligated into pCR3.1-1D8[N18]-TIE-[C18]EGa1-myc/His to obtain pCR3.1-FLAG-StrepII-1D8[N18/C18]EGa1-myc/His. The C-terminal myc/His tag was removed by PCR with LEGA-1 and Stop-XbaI-Rev primers (Supplementary Table 6) to generate pCR3.1-FLAG-StrepII-1D8[N18/][C18]EGa1. The sequences were verified with Fw-CMV and Rv-BGH (Supplementary Table 6).

**Expression and purification of recombinant antibodies**. HEK293 cells were transfected with the appropriated vectors and selected in complete DMEM+500 μg/ml G418 to generate stable cell lines. Recombinant antibodies were purified from conditioned media with HisTrap HP columns or with the (Twin-) Strep-tag purification system (IBA Lifesciences). Endotoxin levels were <0.25 EU/ml as determined by the LAL Endotoxin Kit (Pierce).

**Western blotting**. Samples were separated under reducing conditions on 10–20% Tris-glycine gels and transferred onto nitrocellulose membranes and probed with either anti-c-myc (9E10, cat#ab206486, Abcam) or anti-FLAG (M2, cat#F480, Sigma-Aldrich) mAbs (1 μg/ml), followed by incubation with DyLight800-goat anti-mouse (GAM) IgG (1:5000 dilution) (cat#610-145-121, Rockland Immunochemicals). Visualization of protein bands was performed with the Odyssey® system (LI-COR Biosciences).

**Enzyme-linked immunosorbent assay**. Mouse 4-1BB:hFc chimera (m4-1BB; cat#937-4B-050, R&D Systems) and human EGFR:hFc chimera (hEGFR; cat#ab155726, Abcam) were immobilized (3 μg/ml) on Maxisorp plates (NUNC Brand Products) overnight at 4 °C. After washing and blocking, conditioned media or purified protein solution (1 μg/ml) were added and incubated for 1 h at room temperature. The wells were washed and anti-c-myc mAb or anti-FLAG mAb added (1 μg/ml). After washing, horseradish peroxidase (HRP)-GAM IgG (1:1000 dilution) (cat#A5278, Sigma-Aldrich) or HRP-goat anti-rat (GAR) IgG (1:1000 dilution) (cat#ab97057, Abcam) were added, and the plate was developed using O-phenylenediamine dihydrochloride (OPD). For competition ELISA, m4-1BB (3 μg/ml) was immobilized, and after blocking, m4-1BBL (cat#754402, Biolegend) was added at 4.27 nM for 1 h. Purified antibody solution (4.27 nM) was serially two-fold diluted, added to the wells, and incubated for 1 h. After washing, anti-FLAG mAb (1 μg/ml) was added for 1 h. After washing, HRP-GAM IgG (1:1000 dilution) was added and developed.

**Flow cytometry**. The cells were incubated for 1 h with purified antibodies (6.67 nM), washed, incubated for 30 min with anti-His (BMG-His-1, cat#11922416001, Roche Life Science) anti-c-myc or anti-FLAG mAbs (1 μg/ml), and were detected with a phycoerythrin (PE)-GAM IgG F(ab')₂ antibody (1:200 dilution) (cat#115-116-072, Jackson Immuno Research). The 1D8 IgG (M. Jure-Kunkel, Bristol-Myers Squibb) and cetuximab (Merck KGaA) (6.67 nM) were used as controls, using PE-GAM IgG F(ab')₂ (1:200 dilution) and PE-goat anti-human IgG F(ab')₂ (1:50 dilution) (cat#109-116-097, Jackson Immuno Research), respectively. Samples were analyzed with a MACSQuant Analyzer 10 (Miltenyi Biotec GmbH). For competition studies, purified mouse CD8a⁺ T cells were activated for 48 h with concanavalin-A (5 μg/ml), blocked with human γ-globulin for 5 min, incubated with 10 μg/ml 1D8 IgG for 20 min on ice, and washed with PBS thereafter, while the other samples were left untreated. Then cells were incubated with 2 μg/ml of 1D8 IgG, 1D8[N18], 1D8[N5], or rat IgG[2a] for 20 min on ice and washed. Next, cells that were incubated with 1D8[N18] or 1D8[N5] and one untreated control sample was incubated with rabbit anti-c-myc mAb (1:200 dilution) (A-14; cat#sc789, Santa Cruz Biotechnology). Finally, cells were incubated with either Alexa 647-GAR IgG or donkey anti-rabbit IgG (1:100 dilution) (cat#A-21247 and A-31573, Molecular Probes) together with fluorescein isothiocyanate (FITC)-anti-mouse CD8 mAb (1:500 dilution) (53-6.7; cat#553030, BD Pharmigen) for 20 min. Samples were analyzed with FACSCanto II and FACSort flow cytometers (BD Biosciences). To generate HEK293 cells homogeneously expressing high levels of 4-1BB (HEK293[m4-1BB]-S), HEK293[m4-1BB] cells were stained with PE-anti-mouse CD137 mAb (1:1000 dilution) (17B5; cat#106105, Biolegend) and sorted using a FACSAria II (BD Biosciences).

**Mass spectrometry**. A 2 μl protein sample was desalted using ZipTip® C4 microcolumns (Merck Millipore) and eluted with 0.5 μl sinapinic acid (10 mg/ml in [70:30] Acetonitrile: Trifluoroacetic acid 0.1%) matrix onto a GroundSteel massive 384 target (Bruker Daltonics). An Autoflex III MALDI-TOF/TOF spectrometer (Bruker Daltonics) was used in linear mode with the following settings: 5000–40,000 Th window, linear positive mode, ion source 1: 20 kV, ion source 2: 18.5 kV, lens: 9 kV, pulsed ion extraction of 120 ns, high gating ion suppression up to 1000 Mr. Mass calibration was performed externally with protein 1 standard calibration mixture (Bruker Daltonics). Data acquisition, peak peaking, and subsequent spectra analysis was performed using the FlexControl 3.0 and FlexAnalysis 3.0 software (Bruker Daltonics).

**Size exclusion chromatography with multi-angle light scattering**. Static light scattering experiments were performed at room temperature on a Superdex 200 Increase 10/300 GL column (GE Healthcare) attached in-line to a DAWN-HELEOS light scattering detector and an Optilab rEX differential refractive index detector (Wyatt Technology). The column has an exclusion volume of 8.6 ml, and no absorbance (no aggregated protein) was observed in any of the injections. The column was equilibrated with running buffer (PBS+150 Mm NaCl) and the SEC-MALS system was calibrated with a sample of bovine serum albumin (BSA) at 1 g/l in the same buffer. Then 100 μl samples of the two antibodies 1D8[N18] and 1D8[N/C]EGa1 at 1 g/l in the running buffer were injected into the column at a flow rate of 0.5 ml/min. Data acquisition and analysis were performed using the ASTRA software (Wyatt Technology). The reported molar mass corresponds to the center of the chromatography peaks. After separation of the monomeric species by SEC a second injection in the SEC-MALS system was done at 0.26 g/l. Based on numerous measurements on BSA samples at 1 g/l under the same or similar conditions, we estimate that the experimental error in the molar mass is around 5%.

**Circular dichroism**. Circular dichroism measurements were performed with a Jasco J-810 spectropolarimeter (JASCO). The spectra were recorded on protein samples at 0.2 g/l in PBS using 0.2 cm path length quartz cuvettes at 25 °C. Thermal denaturation curves from 10 to 95 °C were recorded on the same protein samples and cuvette by increasing temperature at a rate of 1 °C/min and measuring the change in ellipticity at 218 nm.

**Small-angle X-ray scattering**. SAXS experiments were performed at the beamline B21 of the Diamond Light Source (Didcot, UK). The proteins were concentrated and prepared at 4 °C prior data collection. Samples of 40 μl of 1D8[N18] and 1D8[N/C]EGa1 at concentrations of 3 and 6 mg/ml were delivered at 4 °C via an in-line Agilent 1200 HPLC system in a Shodex Kw-403 column, using a running buffer composed by 50 mM Tris pH 7.5+150 mM NaCl. The continuously eluting samples were exposed for 300 s in 10 s acquisition blocks using an X-ray wavelength of 1 Å and a sample to detect (Pilatus 2M) distance of 3.9 m. The data covered a momentum transfer range of $0.032 < q < 3.695$ Å$^{-1}$. The frames recorded immediately before elution of the sample were subtracted from the protein scattering profiles. The Scatter software package (www.bioisis.net) was used to analyze data, buffer-subtraction, scaling, merging, and checking possible radiation damage of the samples. The data set corresponding to 1D8[N18] at 3 mg/ml could not be further analyzed due to aggregation. The $R_g$ values were calculated with the Guinier approximation assuming that at very small angles $q < 1.3/R_g$. The maximum particle distribution, $D_{max}$, and the distance distribution were calculated from the scattering pattern with GNOM, and shape estimation was carried out with DAMMIF/DAMMIN, and all these programs are included in the ATSAS package[52]. Interactively generated PDB-based models were made for the two antibodies based in templates obtained with the program RaptorX. Real-space scattering profiles of the models were computed with the program FoXS.

**Kinetic measurements using BLI.** The interactions between immobilized m4-1BB and 1D8 IgG, 1D8$^{N18}$, 1D8$^{N5}$, and 1D8$^{N0}$ were investigated on an Octet RED96 system (Fortebio). Mouse 4-1BB was immobilized onto AR2G biosensors (Fortebio) using 10 μg/ml of m4-1BB in 10 mM acetate buffer at pH 6, over 20 min, to a signal of 1.8 ± 0.4 nm. The kinetics experiment for each antibody used 4 unregenerated m4-1BB-coated biosensors, 2 of which were associated with 4 nM of antibody in kinetics buffer (PBS+0.1% BSA+0.05% Tween 20), and 2 with 2 nM. Association was run for 1 h, followed by 3 h of dissociation in analyte-free kinetics buffer. The acquired sensorgrams were globally fit to a 1:1 model using the Octet Data analysis software. Kinetics experiments were performed at 37 °C while shaken at 1000 rpm. The avidity of the interaction between immobilized hEGFR and analyte 1D8$^{N/C}$EGa1 was investigated similarly. The hEGFR was immobilized onto AR2G biosensors using 3 μg/ml of hEGFR in 10 mM acetate buffer at pH 5. Association and dissociation were both measured for 2 h. To demonstrate the ability of 1D8$^{N/C}$EGa1 to bind both of its antigens in tandem, m4-1BB was immobilized onto AR2G biosensors, and 4 nM of 1D8$^{N/C}$EGa1 or 1D8$^{N18}$ were allowed to associate to 2 biosensors each for 1 h. The biosensors were briefly moved into kinetics buffer for 10 min, after which 1 biosensor loaded with each antibody was moved into 10 nM hEGFR for 1 h while the other biosensor remained in kinetics buffer. Finally, all biosensors were moved into kinetics buffer for 1 h.

**Serum stability.** Purified antibodies were incubated in human serum at 37 °C, for at least 7 days. The binding activity of the sample at 0 h was set as 100% in order to calculate the time corresponding to percentage of decay in binding activity.

**T cell costimulation assays.** Goat anti-hamster IgG (cat#127-005-160, Jackson ImmunoReserach) was pre-coated overnight at 4 °C in 96-well plates (5 μg/ml), and after blocking, anti-CD3 mAb (145-2C11; cat# MO3PU(V100), Immunostep) (1 μg/ml) was added and incubated at 37 °C for 1 h. Mouse CD8a$^+$ T cells were purified from the spleens of C57BL/6 mice using the EasySep™ Mouse CD8a$^+$T Cell Isolation Kit (Stem Cell Technologies). Then purified mouse CD8a$^+$ T cells (2.5 × 10$^5$/well) in complete RPMI+50 μM 2-mercaptoethanol, and purified antibodies at 6.67 nM were added. As a control, purified mouse CD8a$^+$ T cells were cultured alone with the immobilized anti-CD3 mAb (1 μg/ml). After 48 h, cell proliferation was assessed with the CellTiter-Glo luminescent assay (Promega) using a Tecan Infinite F200 plate-reading luminometer, and supernatants were collected and assayed for IFN-γ secretion by ELISA (Diaclone). For viability assays, cells were collected after 72 h, incubated with FITC-Annexin V (Immunostep) and 7-AAD (BD Biosciences), and analyzed with a MACSQuant Analyzer 10. Results are expressed as a mean ± SD from one of at least three separate experiments.

**Antigen-specific T cell costimulation assays.** For studies with purified hEGFR, 96-well plates were pre-coated with goat anti-hamster IgG (5 μg/ml) and hEGFR (5 μg/well). After blocking, anti-CD3 mAb (1 μg/ml) was added and incubated for 1 h at 37 °C, before adding mouse CD8a$^+$ T cells and the purified antibodies (6.67 nM). For studies with cells, NIH/3T3 or 3T3$^{hEGFR}$ target cells were seeded overnight. Next day, target cells were pre-incubated for 30 min on ice with purified antibodies (6.67 nM). Mouse CD8a$^+$ T cells were activated with biotin-anti-CD3 mAb (145-2C11, cat#100303, Biolegend) (100 ng/ml) cross-linked with streptavidin (1:5 molar ratio) and added at 10:1 effector/target ratio. As a control, mouse CD8a$^+$ T cells were cultured alone with plastic immobilized anti-CD3 mAb (1 μg/ml). Cell proliferation and IFN-γ secretion were measured after 48 h, and cell viability after 72 h. Results are expressed as a mean ± SD from one of at least three separate experiments.

**Inhibition of EGFR-mediated cell proliferation and signaling.** A431 cells were seeded in complete DMEM in 96-well plates. After 24 h, medium was replaced by DMEM+1% fetal calf serum (FCS) containing equimolar concentrations (0.19–50 nM) of cetuximab, rituximab (Hoffmann-La Roche Ltd.), 1D8$^{N/C}$EGa1, or 1D8 IgG and incubated for 72 h. Viability was assessed using the CellTiter-Glo luminescent assay. Experiments were performed in triplicates. For EGFR signaling studies, A431cells were starved overnight in DMEM+1% FCS and then incubated for 4 h in serum-free DMEM in the presence of 0.1 μM cetuximab, rituximab, 1D8$^{N/C}$EGa1, or 1D8 IgG, followed by incubation with 25 ng/ml of human EGF (MiltenyiBiotec GmbH) for 5 min. After stimulation, cells were lysed in Laemmli-lysis buffer, separated under reducing conditions on 4–12% Tris-glycine gels, transferred to nitrocellulose membrane, and incubated with the rabbit anti-human phosphor-EGFR (Tyr1068) mAb (1:1000 dilution) (D7A5; cat#3777, Cell Signaling Technology Inc.) followed by incubation with an IRDye800CW-donkey anti-rabbit antibody (1:5000 dilution) (cat#925-32213, LI-COR Biosciences). Simultaneously, anti-β-actin mouse mAb (1:2000 dilution) (8226; cat#ab8226, Abcam) was added as a loading control, followed by IRDye680RD-donkey anti-mouse IgG (1:5000 dilution) (cat#925-68072, LI-COR Biosciences). Visualization and quantitative analysis of protein bands were carried out with the Odyssey system.

**Antibody labeling.** Purified 1D8 IgG, 1D8$^{N18}$, and 4-1BBL were labeled with the Mix-n-Stain CF488A Antibody Labeling Kit, and purified 3H3 IgG (cat#BE0239, BioXCell) and 1D8$^{N/C}$EGa1 were labeled with NIR fluorochrome CF647 using the Mix-n-Stain CF647 Antibody Labeling Kit according to the manufacturer's

recommendations (Biotium). Cetuximab was labeled with NIR fluorochrome Cy5™ Mono NHS Ester (GE Healthcare) according to the manufacturer's recommendations. The degree of labeling (DOL) was determined from the absorption spectrum of the labeled antibody; 1.5/1 and 2.7/1 dye/protein in the case of CF488A- and CF647-labeled proteins, respectively. Cetuximab has a DOL of 2.3/1 dye/protein.

**Live cell receptor clustering imaging.** HEK293$^{m4-1BB}$-S cells were plated onto 35-mm poly-L-lysine pre-coated dishes at a 50% confluence, and after overnight culture, the dishes were assembled in 35-mm diameter chambers (Ibidi GmbH) with 600 μl of dichloromethane and set onto a Leica SP8 3X SMD microscope (Leica Microsystems) under 37 °C and 5% CO$_2$ conditions. The excitation wavelength used was 488 nm from a white light laser (NKT Photonics A/S) with very low power 2–5%, and the detection was from 500 to 550 nm. CF488A-labeled 1D8 IgG, 1D8$^{N18}$, or 4-1BBL were extemporaneously added to the cells at a final concentration of 100 ng/ml, and RICS was performed. Characterization of the microscope point spread function, i.e., the focal volume where the fluorescent dyes diffuse in and out, was done employing 2 μg/ml soluble purified EGFP (Biovision)[34]. The series of images recorded for RICS were of 256 × 256 pixels, with a pixel size of 80 nm, and employing 2-μs dwell time. RICS analysis and diffusion coefficient quantification was done by employing the SIM FCS 4 software (G-SOFT Inc.). Every time trace from the time series was carefully observed to avoid possible artifacts due to dramatic photobleaching, which otherwise would affect the diffusion coefficient quantification. By these means, we avoided employing detrending algorithms that sometimes can bias the analysis. RICS analysis was performed using a moving average (background subtraction) of ten to discard possible artifacts arising from cellular motion and slow-moving particles passing through. The obtained two-dimensional autocorrelation map was fitted to get the diffusion surface map that was represented in three dimension. For the different region of interest (ROI) analyses within the same cell, the corresponding region was drawn employing a square region of 64 × 64 pixels. Selected regions were defined as where there was the presence or absence of receptor clustering. For statistical purposes, each condition was studied on minimum five different cells, and in each cell a minimum of three different ROIs were analyzed. Brightness and contrast of the fluorescence and differential interference contrast were optimized with the ImageJ software. Diffusion values were represented in a whisker and box plot using OriginPro (OriginLab).

**Pharmacokinetics study.** Female CD-1 mice ($n$ = 24/group) received a single i.v. dose of 1D8$^{N18}$ or 1D8$^{N/C}$EGa1 (1 mg/kg), and blood samples from 3 mice per group were collected at 5, 15, 30 min and 1, 3, 6, 24, and 48 h. Female BALB/C mice were injected i.v. ($n$ = 24/group) or i.p. ($n$ = 6/group) with a single dose of 3H3 IgG or 1D8$^{N/C}$EGa1 (1 mg/kg). Serial blood samples were obtained at different time points from 30 min to 264 h. Serum was obtained after centrifugation and stored at −20 °C. Sera were analyzed for antibody concentration by ELISA against immobilized m4-1BB (3 μg/ml). After washing and blocking, sera from different time points were added and incubated for 1 h at room temperature. The wells were washed and HRP-anti-c-myc (1 μg/ml) (cat#ab1326, Abcam), HRP-anti-FLAG mAb (1 μg/ml) (M2; cat#ab49763, Abcam), or HRP-GAR IgG (1:1000 dilution) were added. After washing, the plates were developed using OPD. Pharmacokinetic parameters were calculated using the Prism software (GraphPad Software).

**Molecular imaging in tumor-bearing mice.** A431 cells (1 × 10$^6$) were implanted s. c. into the dorsal space of 6-week-old female Hsd:athymic Nude-Foxn1nu mice. Tumor growth was monitored two times a week by measuring the diameter of the tumors with a calliper, and tumor volumes were calculated according to the formula: Volume = ($D$ × d2/2), where $D$ is the longest axis of the tumor and $d$ is the shortest of a prolate ellipse. When tumor volume reached about 0.180 cm$^3$, mice were randomly allocated to different treatment groups ($n$ = 3/group) and i.v. injected with PBS or NIR-labeled antibody solution (1 mg/ml) in PBS. Mice were imaged under anesthesia at 24 h under the IVIS Spectrum CT in vivo imaging system (Xenogen) at the indicated wavelengths and were analyzed using the Living Image 3.2 software (PerkinElmer). The images were analyzed by identifying 3 ROIs within the tumor and from surrounding regions (normal tissue). The T/N ratio was calculated by dividing the mean values of the identified ROIs. Fluorescence intensity of all the images are reported as photons per second per centimeter squared per steradian (p/s/cm$^2$/sr).

**Therapeutic studies.** CT26$^{hEGFR}$ cells (1.5 × 10$^6$) were implanted s.c. into the dorsal space of 6-week-old female BALB/c mice. Tumor growth was monitored by calliper measurements three times a week, and when tumors reached approximately 0.4 cm in diameter (usually from 7 to 10 days), mice were randomized to receive treatment ($n$ = 5 or 6/group). Measurements were conducted in a random order by the investigator who was blinded to the treatment assignment. Mice were treated every other day with three i.p. injections of PBS, anti-4-1BB antibodies, or control antibodies (4 mg/Kg). Mice were euthanized when tumor size reached a diameter of 1.5 cm any dimension, when tumors ulcerated, or at any sign of mouse distress. To study the long-lasting systemic immune-mediated response, surviving

mice ($n = 5$) were re-challenged with CT26$^{mock}$ cells ($1.5 \times 10^6$) s.c. in the contralateral left flank 50 days following treatment with 3H3 IgG or 1D8$^{N/C}$EGa1. Cured and naive mice were followed for an additional 190 days after reinoculation.

**Immunohistochemistry.** CT26$^{hEGFR}$ tumors from different treatment groups were collected after 13 days of implantation and fixed in 10% neutral buffered formalin (Sigma-Aldrich) for 48 h. Then, after extensive washing in PBS, tissues were embedded in paraffin, cut at 3 μm, mounted in Superfrost®plus slides, and dried overnight. Slides were deparaffined in xylene and re-hydrated through a series of graded ethanol washes, ending in a final rinse in pure water. Slides were incubated with a rat monoclonal anti-CD8a (1:200 dilution) (OTO94A; Monoclonal Antibodies Core Unit CNIO) followed by a rabbit anti-rat secondary antibody (cat#ab6703, Abcam) and a visualization system (Novolink Polymer anti-Rabbit, Leica) conjugated with HRP. Nuclei were counterstained with Harris' hematoxylin. Positive control sections known to be primary antibody positive were included for each staining run. Whole digital slides were acquired with a slide scanner (AxioScan Z1, Zeiss), and total versus positive cells were automatically quantified (AxioVision 4.6 software package, Zeiss).

**Toxicity studies.** Three-month-old female C57BL/6 received a weekly i.p. dose of 3H3 IgG or 1D8$^{N/C}$EGa1 (6 mg/kg) for 3 weeks. Mice were anesthetized and bled on days 0, 7, 14, and 21. To obtain mouse serum, blood was incubated in BD microtainer SST tubes (BD Biosciences), followed by centrifugation. Serum was stored at −20 °C until use. One week after the last dose of antibodies, mice were euthanized and the liver, spleens, lungs, and pancreas were surgically removed, weighted, and fixed in 10% paraformaldehyde for 48 h. Then fixed tissues were washed and embedded in paraffin. Tissue sections (5 μm) were stained with hematoxylin and eosin, Sirius red, or Masson's trichrome (Sigma-Aldrich) for collagen staining. Lymphocyte infiltration in the liver was quantified using the ImageJ software. A similar study was performed in 2-month-old female BALB/c mice, but in this case the 1D8$^{N/C}$EGa1 was injected i.p. every 3/4 days (6 mg/kg) for 3 weeks. The 3H3 IgG was administered with the same regimen: a weekly i.p. injection (6 mg/kg) for 3 weeks.

**Luminex assay.** Blood was collected from the treated mice on the indicated days, and the levels of inflammatory cytokines (IFN-γ, IL-6, TNF-α) in serum samples were determined using a Luminex Milliplex Magnetic Bead Kit (Merck Millipore).

**Statistical analysis.** Statistical analysis was performed using the GraphPad Prism Software version 5.0. All the in vitro experiments were done in triplicates and values are presented as mean ± SD from one of at least three separate experiments. Significant differences ($P$ value) were discriminated by applying a two-tailed, unpaired Student's $t$ test assuming a normal distribution with *$P \leq 0.05$, **$P \leq 0.01$, ***$P \leq 0.001$. To assess differences in tumor growth, tumor diameter for individual mice in each treatment group are presented. Survival curves for the different treatment groups were created using the Kaplan–Meier method, and two or more survival curves were analyzed using log-rank (Mantel–Cox) test. For the other in vivo studies, data were presented using a scatter plot as mean ± SD, and $P$ values were determined by unpaired Student's $t$ test to assess the differences between treatment groups.

## Data availability
The authors declare that the data supporting this study are available within the paper and its Supplementary Information File. All other data are available from the authors upon reasonable request.

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

## Acknowledgments

We thank M. Jure-Kunkel, M. Rescigno and A. Villalobo for reagents, M.G. Gonzalez Bueno and B. Acosta (IIBm) for technical support, and the staff of beamline B21at Diamond Light Source (Didcot, UK) for their excellent technical assistance. This study was supported by grants from the European Union [IACT Project (602262), iNEXT Project (1676) and a Marie Curie Career Integration Grant (PCIG13-GA-2013-618914)], the Ministerio de Economía y Competitividad (CTQ2017-83810-R, RTC-2016-5118-1, SAF2017-83267-C2-1-R, SAF2017-89437-P, and PTQ-16-08340), the Fondo de Investigación Sanitaria/Instituto de Salud Carlos III (PI16/00357 and PI16/00895), the UK Research and Innovation (18130023), and the Danish Council for Independent Research (DFF-6110-0053). The CIC bioGUNE is a Severo Ochoa Center of Excellence (Ministerio de Economia y Competitividad award SEV-2016-0644). This study was also funded by FEDER.

## Author contributions

L.A.-V. and M.C. conceived and supervised the study. M.C., S.L.H., I.G.M., R.N., M.Z., G. P.-C., A.E.-L., N.M., A.T.-G., A.M.C., K.M., E.C., N.N.-P., M.A.A., S.L., and J.M.-T. designed and performed most of the experiments. M.C., S.L.H., I.G.M., J.M.-T., I.M., F.J. B., J.B.S., J.M.Z., L.S., and L.A.-V. analyzed and discussed the data and wrote the manuscript. All authors edited the manuscript.

## Additional information

**Competing interests:** M.C., M.Z., and A.E.-L. are all employees of Leadartis. L.A.-V. and L.S. are co-founders of Leadartis. The remaining authors declare no competing interests.

