## [Peer Review File · Nature Communications]

Reviewer #1 (Remarks to the Author):

The manuscript of Luis Alvarez-Vallina describes the development of a double trimeric molecule, targeting both 4-1BB and EGFR to treat certain cancers. It is a proof-of concept study, because it targets 4-1BB in the mouse, and still needs to be translated to the clinic. The major claims are that the trimeric binding to 4-1BB leads to more T cell activation and proliferation, with less cytotoxicity as was seen with the human IgG4 antibody urelumab in a phase II trial.

In my opinion this approach is novel, although bispecific molecules combining 4-1BB targeting and tumor targeting have been tried before or are in development.

My major concern about this manuscript is the lack of information often provided. The authors mention that 4-1BB is also expressed on macrophages, activated B cells, and dendritic cells, but they do not show or mention what the effects are of the trimeric molecule on these cells. The major toxicity seen in patients was associated with inflammation in the liver, but in the mouse model the authors show in the end they ignore the fact that the trimeric molecule has a much shorter half-life than the comparator, a full anti 4-1BB antibody, which explains the difference in the adverse effects (figure 6). There for, in figure 4 the PK/PD properties of the full length 4-1BB antibody should have been included, and some proper modelling should have led to a treatment regimen to obtain similar levels of both molecules. Now the dosing is just once a week for 3 weeks with 6mg/kg. In the tumor model in figure 4 the mice were injected every 2 days.

In my opinion, this is an interesting paper but would need many improvements to be interesting to others. More concerns:

- legends are often hard to read/interpret
- sometimes small lettering in figures, hard to read.
- why does the trimeric molecule contain a his-tag and the EGFR molecule a flag-strep tag?
- Why were the binding studies performed with BLI and not with SPR (more sensitive)?
- Please explain better what RICS and SAXS are
- It is puzzling how m4-1BB can have an inhibitory effect on T cell IFN γ secretion in figure 2.
- in figure 4B the double trimeric molecule is compared to a full length 4-1BB mAb. It should show a EGFR mAb, to show imaging of an EGFR positive tumor.
- What is the difference between the last plot in figure 5A and 5B? the treatment seems exactly the same, but in 5A 3 out of 5 mice respond, and in 5B 5 out of 5?

In general, this is an interesting study, but I would like to see a comparison with a bispecific molecule to 4-1BB and EGFR, other targets besides EGFR (Her2? CD20?). More tumor models than just a sc model with human EGFR transduced mouse model, where the human EGFR is a xeno-molecule for the mouse. And better tox studies, with comparable in vivo levels of the molecules.

Reviewer #2 (Remarks to the Author):

This study addresses an important challenge in using 4-1BB costimulation to promote tumor immunity -therapeutic index, namely the severe toxicity associated with systemic administration of antibody-based 4-1BB ligands most likely due to the nonspecific activation of autoreactive T cells. To that end this study describes an antibody-based 4-1BB ligand that is targeted to the tumor. Several previous studies have described various approaches to target 4-1BB and other immune modulatory ligands to tumor lesions in situ. If or which approach is best suited to clinical application is yet to be determined. This study describes a novel approach whereby a trivalent 4-1BB ligand consisting of scFv domains is conjugated to a targeting ligand consisting of a trimeric single-domain EGFR antibody. The use of a trimeric 4-1BB ligand design distinguishes this approach from previously published strategies and may represent an advantage since its target, the cell surface expressed 4-1BB receptor and its natural ligands form trimers during their interaction. But this has not been actually shown.

On the negative side, the construction of the trivalent 4-1BB-EGFR fusion protein is complex and its high molecular weight could significantly impede its intratumoral penetration (see below). In the first part of the study the investigators have gone to a great length to characterize the trimeric fusion protein biochemically and functionally in vitro, necessary and appropriate. In view of the limitations of this reviewer the evaluation of this manuscript will focus primarily on the in vivo studies. The three important questions are targeting, tumor inhibition, and toxicity.

1. Targeting. Study provides biochemical evidence that in contrast to 4-1BB Ab the trimeric fusion protein exhibits preferential tumor accumulation (Fig. 4C). This formally proves the point.

Intratumoral penetration of macromolecules above 40 Kdal is becoming limiting. It is somewhat surprising and reassuring that the targeting experiment (and indirectly the efficacy w/o toxicity experiments) suggest that there is sufficient tumor accumulation that can translate into a relevant biological response, i.e. tumor inhibition. But this study has not evaluated the efficiency of targeting, to what extent the large fusion protein is hindered from effective penetration. In fact that should be studied in spontaneous tumor formation models that recapitulate the architecture of the tumor and its stroma, not bolus-infected tumor cells.

2. Tumor inhibition. The trimeric fusion protein was comparably effective to 4-1BB Abs (Fig. 5A & B) but appears to have elicited more robust long-term protective immunity (Fig. 5C). Dose titration of targeted and nontargeted ligands could have uncovered additional advantages of the trimeric fusion protein both in term of efficacy and reduced concerns of toxicity, and offset concerns of cost of goods.

Absence of immunological studies corroborating the tumor inhibition experiments is an across the board weakness of this study.

3. Toxicity. This is the most important aspect of this study and the data presented in Fig. 6 showing no evidence of immune toxicity compared to the nontargeted 4-1BB Ab are compelling.

Overall this is a proof-of-concept study of a new approach to target immune modulation to the tumor as a way to reduce associated toxicities, arguably an important challenge in cancer immunotherapy. Future studies need to establish the generality and feasibility to generate such trimeric fusion proteins against endogenous targets (other than human EGFR), demonstrate its benefits in autochthonous tumor models and superiority to alternative strategies.

Point-by-point response to the reviewer's comments

Reviewer comments:

Reviewer #1 (Remarks to the Author):

The manuscript of Luis Alvarez-Vallina describes the development of a double trimeric molecule, targeting both 4-1BB and EGFR to treat certain cancers. It is a proof-of concept study, because it targets 4-1BB in the mouse, and still needs to be translated to the clinic. The major claims are that the trimeric binding to 4-1BB leads to more T cell activation and proliferation, with less cytotoxicity as was seen with the human IgG4 antibody urelumab in a phase II trial.

In my opinion this approach is novel, although bispecific molecules combining 4-1BB targeting and tumor targeting have been tried before or are in development. My major concern about this manuscript is the lack of information often provided. The authors mention that 4-1BB is also expressed on macrophages, activated B cells, and dendritic cells, but they do not show or mention what the effects are of the trimeric molecule on these cells. The major toxicity seen in patients was associated with inflammation in the liver, but in the mouse model the authors show in the end they ignore the fact that the trimeric molecule has a much shorter half-life than the comparator, a full anti 4-1BB antibody, which explains the difference in the adverse effects (figure 6). There for, in figure 4 the PK/PD properties of the full length 4-1BB antibody should have been included, and some proper modelling should have led to a treatment regimen to obtain similar levels of both molecules. Now the dosing is just once a week for 3 weeks with 6mg/kg. In the tumor model in figure 4 the mice were injected every 2 days.

Response: Based on the Reviewer's recommendation, in the modified manuscript we have included in Figure 4 the PK/PD properties of the full-length anti-4-1BB mAb 3H3 IgG after i.v. and i.p. injection, panels B and C respectively (see also the new supplementary Table 4 and 5).

In this work, we have used full-length anti-4-1BB agonistic antibodies (3H3 and 1D8) as comparators, therefore we have used protocols that have been validated with those IgG-based anti-4-1BB antibodies, both for anti-tumor efficacy and in toxicity studies.

- In the anti-tumor efficacy studies we have used a therapeutic regimen with 3 i.p. injections every other day, and even knowing that this protocol favors IgG-based molecules with a longer half-life, we have decided to maintain it, to obtain actual comparative anti-tumor efficacy data.
- However, to rule out that the absence of toxicity in 1D8^{N/C}EGa1-treated mice is due to differences in half-life of the 1D8^{N/C}EGa1 trimerbody, following the reviewer's recommendations in the revised manuscript we have included an additional toxicity study in which we have increased the dosage of the 1D8^{N/C}EGa1 trimerbody, from 3 to 6 injections i.p. injections, at 3-day intervals, 3 weeks. The results, included in the new supplementary Figure 19, clearly demonstrated that even with this therapeutic regimen the tumor-targeted 4-1BB agonistic trimerbody is not associated with adverse effects as observed in mice treated with IgG-based anti-4-1BB antibodies.

In my opinion, this is an interesting paper but would need many improvements to be interesting to others. More concerns:

- legends are often hard to read/interpret

Response: following the reviewer's recommendation the figure legends have been modified to facilitate reading and interpretation.

- sometimes small lettering in figures, hard to read.

Response: the letter size have been revised and modified to conform to the guidelines of the journal.

- why does the trimeric molecule contain a his-tag and the EGFR molecule a flag-strep tag?

Response: the first generation of 4-1BB-agonistic trimerbodies was designed with a C-terminal polyhistidine-tag. The purification of recombinant proteins from mammalian cell culture using immobilized metal-affinity chromatography (IMAC) is complex, and recent data in the literature and our own experience with other constructs indicate that StrepTactin affinity is more suitable for mammalian protein expression systems. Therefore, in an attempt to improve the performance (yield and purity) in the purification process from conditioned medium of transfected mammalian cells we decided to incorporate an N-terminal strep-flag tag in the EGFR-targeted 4-1BB-agonistic trimerbody. This change resulted in a significant improvement from 1 mg/L to 5 mg/L.

- Why were the binding studies performed with BLI and not with SPR (more sensitive)?

Response: For both BLI and SPR, sensitivity is related to the molecular weight of the analytes under investigation, and in our case all analyte molecules were sufficiently large to give a robust signal (in our experience, 15 kDa single domain antibodies can be managed using BLI, and the smallest analyte in the present study is 103 kDa) Additionally, BLI generally has a higher throughput than SPR; our RED96 platform uses 8 parallel biosensors, which facilitated our investigation of the kinetics of 9 interactions at different concentrations.

- Please explain better what RICS and SAXS are

Response: additional information has been included in the revised manuscript.

- It is puzzling how m4-1BB can have an inhibitory effect on T cell IFN γ secretion in figure 2.

Response: Inhibitory effects of soluble 4-1BBL on the proliferation of CD3-stimulated T cells have been published (*Rabu et al. Production of recombinant human trimeric CD137L (4-1BBL). Cross-linking is essential to its T cell co-stimulation activity. J Biol Chem. 2005, 50:41472*), but mainly in PBMC, not in purified T cells. We agree with the

reviewer that it is an unexpected result and we attribute it to the fact that the purified recombinant murine 4-1BB ligand used in the study is commercial (Biolegend 754406), and although we have made a preliminary functional and structural characterization (Supplementary Figure 3), due to the limited amount available we can not exclude some integrity problems in the protein preparation, as well as the presence of potential contaminants.

- in figure 4B the double trimeric molecule is compared to a full length 4-1BB mAb. It should show a EGFR mAb, to show imaging of an EGFR positive tumor.

Response: Based on the Reviewer's recommendation, in the modified manuscript we have included the *in vivo* imaging of an EGFR-positive tumor with a cyanine 5 (CyTM5)-labeled full-length anti-human EGFR antibody cetuximab (Figure 4, panel D, left). CFTM 647 is a cyanine-based far-red fluorescent dye spectrally similar to CyTM5 and Alexa Fluor® 647 (CF647 Abs/Em Max: 650/665 nm; Cy5 Abs/Em Max: 649/666 nm). CF647 also has comparable brightness and photostability and is a direct replacement for Cy5. The main difference is the manufacturer: CF is a trademark of Biotium and Cy is a trademark of GE Healthcare. We use CFTM 647 and CyTM 5 interchangeably, choosing for each labeling experiment the dye that most preserves the antibody binding activity as assessed by ELISA. CyTM5, for example, was convenient for cetuximab labeling, but interfered with the functionality of 1D8^{N/C}EGa1 trimerbody.

- What is the difference between the last plot in figure 5A and 5B? the treatment seems exactly the same, but in 5A 3 out of 5 mice respond, and in 5B 5 out of 5?

Response: There are no differences; these are two different groups of mice treated in the same conditions with the 1D8^{N/C}EGa1 trimerbody. In the experimental cancer model used in this work, the implantable CT26 colon carcinoma in fully immunocompetent BALB/c mice, there is a percentage of mice that despite an initial response to treatment progress. Complete tumor regression is achieved in over 80% of mice after 4-1BB mAb administration (see. *Escuin-Ordinas H et al., J Immunother Cancer. 2013, 1:14*). Please note that in the same figure the tumor regression rate is different in mice treated with two IgG-based 4-1BB agonistic mAbs: 1D8 (83.3%) and 3H3 (100%).

In general, this is an interesting study, but I would like to see a comparison with a bispecific molecule to 4-1BB and EGFR, other targets besides EGFR (Her2? CD20?). More tumor models than just a sc model with human EGFR transduced mouse model, where the human EGFR is a xeno-molecule for the mouse. And better tox studies, with comparable in vivo levels of the molecules.

Response: Obviously, the modular design of the 1D8^{N/C}EGa1 molecule is amenable to the use of different tumor targeting domains easily interchanged. In the mid term, we envision a family of trimerbodies directed against different selected tumor-associated antigens (TAAs) to treat not only colorectal cancer but also other solid tumors. Currently, it is not the scope of this work: here, we present the proof of concept demonstrating the therapeutic effect (and absence of toxicity) of bispecific tumor-targeted 4-1BB-agonistic trimerbodies. We think that experiments with bispecific 4-1BB-agonistic trimerbodies

targeting other TAAs would provide valuable confirmatory data but not essential support for the conclusions of this work.

We agree that s.c. models are not the best, but unfortunately they are the only we can use to test the 1D8^{N/C}EGa1 molecule, due to its hybrid nature. Please, take into account that the EGa1 V_{HH} targeting domain recognizes human EGFR. This is why we implanted mouse CT26 colon carcinoma cells genetically engineered to express human EGFR (CT26^{EGFR}) to check the therapeutic effect *in vivo*. A spontaneous colon carcinoma model would express mouse EGFR, and the tumor targeting effect would be lost. Up to date, no anti-mouse EGFR binding domain is available in our lab. Another issue is tumor homing experiments: subcutaneously injected tumor cells are amenable to optical molecular imaging techniques based on fluorescence or luminescence. However, orthotopic tumors would probably be too deep to be imaged with the available equipment.

In the revised manuscript a new toxicity study was conducted in which 1D8^{N/C}EGa was administered (6 mg/kg) i.p. every 3 or 4 days, for 3 weeks, for a total of six doses. These more frequent injections were intended to maintain circulatory levels of 1D8^{N/C}EGa comparable to those of the longer-circulating 3H3 IgG, which was injected as before with 6 mg/kg i.p injections once weekly for 3 weeks, in order to investigate whether 1D8^{N/C}EGa1's shorter half-life might be responsible for its lack of toxicity. The escalated 1D8^{N/C}EGa1 regimen did not induce splenomegaly or hepatomegaly, nor significant histological alterations, while the standard 3H3 IgG regimen induced alterations similar to those observed in 3H3 IgG-treated C57BL/6 mice (Supplementary Fig. 19).

Reviewer #2 (Remarks to the Author):

This study addresses an important challenge in using 4-1BB costimulation to promote tumor immunity -therapeutic index, namely the severe toxicity associated with systemic administration of antibody-based 4-1BB ligands most likely due to the nonspecific activation of autoreactive T cells. To that end this study describes an antibody-based 4-1BB ligand that is targeted to the tumor. Several previous studies have described various approaches to target 4-1BB and other immune modulatory ligands to tumor lesions *in situ*. If or which approach is best suited to clinical application is yet to be determined. This study describes a novel approach whereby a trivalent 4-1BB ligand consisting of scFv domains is conjugated to a targeting ligand consisting of a trimeric single-domain EGFR antibody. The use of a trimeric 4-1BB ligand design distinguishes this approach from previously published strategies and may represent an advantage since its target, the cell surface expressed 4-1BB receptor and its natural ligands form trimers during their interaction. But this has not been actually shown.

On the negative side, the construction of the trivalent 4-1BB-EGFR fusion protein is complex and its high molecular weight could significantly impede its intratumoral penetration (see below). In the first part of the study the investigators have gone to a great length to characterize the trimeric fusion protein biochemically and functionally *in vitro*, necessary and appropriate. In view of the limitations of this reviewer the evaluation of this manuscript will focus primarily on the *in vivo* studies. The three important questions are targeting, tumor inhibition, and toxicity.

1. Targeting. Study provides biochemical evidence that in contrast to 4-1BB Ab the

trimeric fusion protein exhibits preferential tumor accumulation (Fig. 4C). This formally proves the point.

Intratumoral penetration of macromolecules above 40 Kdal is becoming limiting. It is somewhat surprising and reassuring that the targeting experiment (and indirectly the efficacy w/o toxicity experiments) suggest that there is sufficient tumor accumulation that can translate into a relevant biological response, i.e. tumor inhibition. But this study has not evaluated the efficiency of targeting, to what extent the large fusion protein is hindered from effective penetration. In fact that should be studied in spontaneous tumor formation models that recapitulate the architecture of the tumor and its stroma, not bolus-infected tumor cells.

Response: Indeed, tumor penetration is a concern when using large macromolecules for targeting. However, we do not consider our molecule to be so large: the EGFR-targeted 4-1BB-agonistic trimerbody calculated molecular weight is 158.7 kDa, very close to the 152 kDa of cetuximab (or any other IgG, in fact). The therapeutic effect of cetuximab was demonstrated in numerous clinical trials, which prompted its FDA's approval in 2004, and apparently its size did not create doubts about sufficient tumor accumulation. Trastuzumab, another IgG1 with approximately the same size, in the market for the last 20 years, was the first therapeutic antibody targeting a solid tumor. Currently, more than a dozen antibodies or antibody-drug conjugates comprising an IgG1 or IgG2 directed against a solid tumor-associated antigen have been approved or are in phase III clinical trials. Would you consider the success of such therapeutic IgGs to be "somewhat surprising" due to their size? Obviously, the biological response of FDA-approved antibodies is clinically relevant, and we assume that enough antibody accumulates in the tumor to provoke it. We assume that tumor accumulation would be better for a molecule smaller than an IgG (and we have worked for decades with antibody fragments using this argument), but the fact is that IgGs work. And the same reasoning can be applied to the 1D8^{N/C}EGa1 trimerbody: sizes are equivalent, and 1D8^{N/C}EGa1 also works *in vivo*, therefore tumor penetration should not be a concern. In the revised manuscript we have included the *in vivo* imaging of an EGFR positive tumors with a NIR-labeled anti-EGFR antibody cetuximab, and the 1D8^{N/C}EGa1 trimerbody showed high tumor localization with a tumor to normal tissue (T/N) ratio of 4.85 ± 0.13 , as compared to that of cetuximab (2.54 ± 0.34) ($P \leq 0.01$), and 3H3 IgG (1.29 ± 0.06) ($P \leq 0.0001$) (Fig. 4D and E).

Regarding spontaneous tumor models, it is true that offer numerous advantages, but unfortunately we can not use them to test our trimerbody, since the EGa1 V_{HH} antibody recognizes human EGFR. This is why we implanted mouse CT26 colon carcinoma cells genetically engineered to express human EGFR (CT26^{EGFR}) to check the therapeutic effect *in vivo*. A spontaneous colon carcinoma model would express mouse EGFR, and the tumor targeting effect would be lost. Another issue is tumor homing experiments: subcutaneously injected tumor cells are amenable to optical molecular imaging techniques based on fluorescence or luminescence. However, orthotopic tumors would probably be too deep to be imaged with the available equipment.

2. Tumor inhibition. The trimeric fusion protein was comparably effective to 4-1BB Abs (Fig. 5A & B) but appears to have elicited more robust long-term protective immunity (Fig. 5C). Dose titration of targeted and nontargeted ligands could have uncovered additional advantages of the trimeric fusion protein both in term of efficacy and reduced concerns of toxicity, and offset concerns of cost of goods.

Response: We think that experiments suggested by the reviewer would provide valuable information but not essential support for the conclusions of this work. Here, we present the proof of concept demonstrating for the first time the therapeutic effect and absence of toxicity of bispecific tumor-targeted 4-1BB-agonistic trimerbodies

Absence of immunological studies corroborating the tumor inhibition experiments is an across the board weakness of this study.

Response: In the modified manuscript we have included the results of a new anti-tumor efficacy study (Supplementary Fig.15) to characterize the percentage of tumor infiltrating CD8⁺ T lymphocytes (TILs) (Fig. 5D and S16). The percentage of CD8⁺ TILs was an order of magnitude higher in 3H3 IgG- and 1D8^{N/C}EGa1-treated mice (19.48 ± 1.28 and 16.25 ± 1.41 , respectively) compared to the PBS-treated mice (1.42 ± 0.36) (Fig. 5D and S16), indicating a clear recruitment of CD8⁺ T lymphocytes in the antibody-treated mice. The mechanism of action of IgG-based 4-1BB-agonistic antibodies has been interpreted as a consequence of enhanced antitumor CD8⁺ T cell responses, and as the percentage of infiltrating CD8⁺ T cells is similar in 3H3 IgG- and 1D8^{N/C}EGa1-treated mice, it is likely that both IgG-based anti-4-1BB antibodies and tumor-targeted 4-1BB agonist trimerbodies act in a mechanistically equivalent manner.

3. Toxicity. This is the most important aspect of this study and the data presented in Fig. 6 showing no evidence of immune toxicity compared to the nontargeted 4-1BB Ab are compelling.

Response: In the revised manuscript a new toxicity study was conducted in which 1D8^{N/C}EGa was administered (6 mg/kg) i.p. every 3 or 4 days, for 3 weeks, for a total of six doses. These more frequent injections were intended to maintain circulatory levels of 1D8^{N/C}EGa comparable to those of the longer-circulating 3H3 IgG, which was injected as before with 6 mg/kg i.p injections once weekly for 3 weeks, in order to investigate whether 1D8^{N/C}EGa1's shorter half-life might be responsible for its lack of toxicity. The escalated 1D8^{N/C}EGa1 regimen did not induce splenomegaly or hepatomegaly, nor significant histological alterations, while the standard 3H3 IgG regimen induced alterations similar to those observed in 3H3 IgG-treated C57BL/6 mice (Supplementary Fig. 19).

Overall this is a proof-of-concept study of a new approach to target immune modulation to the tumor as a way to reduce associated toxicities, arguably an important challenge in cancer immunotherapy. Future studies need to establish the generality and feasibility to generate such trimeric fusion proteins against endogenous targets (other than human EGFR), demonstrate its benefits in autochthonous tumor models and superiority to alternative strategies.

Response: we agree with the reviewer, and in fact studies are being initiated to generate tumor-targeted 4-1BB agonist trimerbodies recognizing murine tumor-associated antigens.

Reviewer #1 (Remarks to the Author):

The manuscript has significantly improved and most of my concerns were answered, so I don't have any further comments.

Reviewer #2 (Remarks to the Author):

None

Point-by-point response to the reviewer's comments

We would like to thank the reviewers for the helpful comments and suggestions